# The effects of a second pregnancy on women's brain structure and function

M. Straathof [1] ✉, S. Halmans [1], P. J. W. Pouwels [2], E. A. Crone[3,4] &
E. Hoekzema [1] ✉

While we have previously shown that a first pregnancy changes women's brain structure and resting-state brain activity, it is currently unknown how a woman's brain is transformed when she undergoes another pregnancy. Therefore, we performed a prospective pre-conception cohort study involving 110 women, including women who became pregnant of their second (PRG2) or first child (PRG1) and nulliparous women. Multimodal MRI data were acquired and differential changes between PRG2 and PRG1 were observed in grey matter volume, white matter tracts and functional neural network organization. Together, these results show similar but less pronounced structural and functional changes in the default mode and frontoparietal network in PRG2, suggesting a primary adaptation of these networks in first-time mothers that is further fine-tuned across a second pregnancy. Furthermore, stronger alterations were found in PRG2 in the dorsal attention and somatomotor network including the corticospinal tract, pointing to an enhanced plasticity within these externally-oriented networks. Neurostructural changes in both groups related to mother-infant attachment and peripartum depression. These findings show that a second pregnancy uniquely changes a woman's brain, entailing both convergent and distinct neural transformations.

Pregnancy represents a monumental phase in many women's lives, orchestrated by unparalleled physiological and neuroendocrine changes that influence all major bodily systems. We have previously shown that reproduction is associated with widespread changes in women's brain structure[1,2], which has since been replicated in various other studies across the world[3–5]. Additionally, we have shown that pregnancy also affects functional neural network organization in the default mode network[2]. While research is beginning to uncover pregnancy-induced brain plasticity, all studies so far have focused solely on first-time mothers, leaving it unclear whether similar brain changes occur as a result of subsequent pregnancies.

Animal studies point to an association between subsequent reproductive experiences and neuroanatomical variations in the brain. Hippocampal dendritic morphology, neurogenesis and protein levels have been shown to differ between primiparous and multiparous female rodents[6–8]. Additionally, parity has lasting effects on synaptic plasticity and inflammation in the hippocampus and the response to ovarian hormones in middle-aged rodents[9–12]. While these findings provide evidence for structural variations in the rodent brain on the basis of parity, research on this topic in human mothers is still in its infancy.

Recent studies have suggested that parity may involve long-lasting neural changes in human mothers measured beyond the postpartum period. In middle-aged women, parity has been linked to brain age, with multiparous women having younger-looking brains compared to primiparous and nulliparous women[13,14]. Additionally,

[1]Pregnancy Brain Lab, Amsterdam University Medical Center (UMC), Location University of Amsterdam, Department of Psychiatry, Amsterdam Neuroscience, Amsterdam Reproduction and Development, Amsterdam, The Netherlands. [2]Amsterdam University Medical Center (UMC), Location Vrije Universiteit Amsterdam, Department of Radiology and Nuclear Medicine, Amsterdam Neuroscience, Amsterdam, The Netherlands. [3]Brain and Development Research Centre, Leiden Institute for Brain and Cognition, Leiden University, Leiden, The Netherlands. [4]Erasmus School of Social and Behavioural Sciences, Erasmus University Rotterdam, Rotterdam, The Netherlands. ✉e-mail: m.straathof@amsterdamumc.nl; e.a.hoekzema@amsterdamumc.nl

parity has been associated with cortical thickness, functional connectivity and gray matter volume in women during late life[15–17]. Parity may also modulate the risk for and effects of various brain disorders that may occur later in life, such as Alzheimer's Disease, stroke and traumatic brain injury[18–21]. Furthermore, links between parity and cognition have been shown[22], characterized by a U-shaped effect of parity on cognition, with the highest scores on the Mini Mental State Examination in women with one to four pregnancies, compared to nulliparous women and women with grand parity[23]. Although these findings provide evidence for long-lasting differences in neural anatomy in relation to multiple childbirths in human mothers, it remains unclear whether a first and second pregnancy differentially affect the human brain.

Therefore, we performed a prospective pre-conception cohort study involving second-time mothers, first-time mothers and nulliparous control women to investigate whether a second pregnancy is associated with structural and functional changes in the brain. We acquired multimodal 3 T magnetic resonance imaging (MRI) data, involving high-resolution anatomical MRI, resting-state functional MRI, diffusion-weighted MRI and magnetic resonance spectroscopy (MRS), before pregnancy and in the early and late postpartum period in 30 women undergoing a second pregnancy (PRG2; becoming multiparous between the scans) and in 40 women undergoing a first pregnancy (PRG1; becoming primiparous between the scans). We also included a control group of 40 nulliparous control women (CTR), who

underwent MRI scanning at a similar time interval. We assessed the differential effects of a second pregnancy on brain structure, functional neural networks, white matter organization and neural metabolite concentrations. This study demonstrates that a second pregnancy changes a woman's brain and uniquely impacts its gray matter structure, neural network organization and white matter tracts.

## Results

### Vertex-wise volumetric brain changes

To investigate anatomical brain changes across a first and second pregnancy compared to control women, we performed vertex-wise longitudinal analyses of cortical volume, thickness and surface area in Freesurfer. Generalized linear models (GLMs) comparing the volume change from pre-pregnancy (PRE) to the early postpartum period (POST) between multiparous and control women demonstrated widespread volumetric decreases across a second pregnancy (Fig. 1a, b). The median percentage volume decrease across these significant vertices in the left and right hemisphere was 2.8%. Similarly, women who were pregnant for the first time showed significant volumetric decreases from PRE to POST compared to control women (Fig. 1c, d). The areas of significant change covered a 79% larger part of the brain than in second-time mothers, and the median percentage of change was 3.1% within these significant vertices. Effect sizes for these volumetric decreases across a first and second pregnancy were large (Cohen's $D > 1.0$; Supplementary Table 1). In both groups, there were

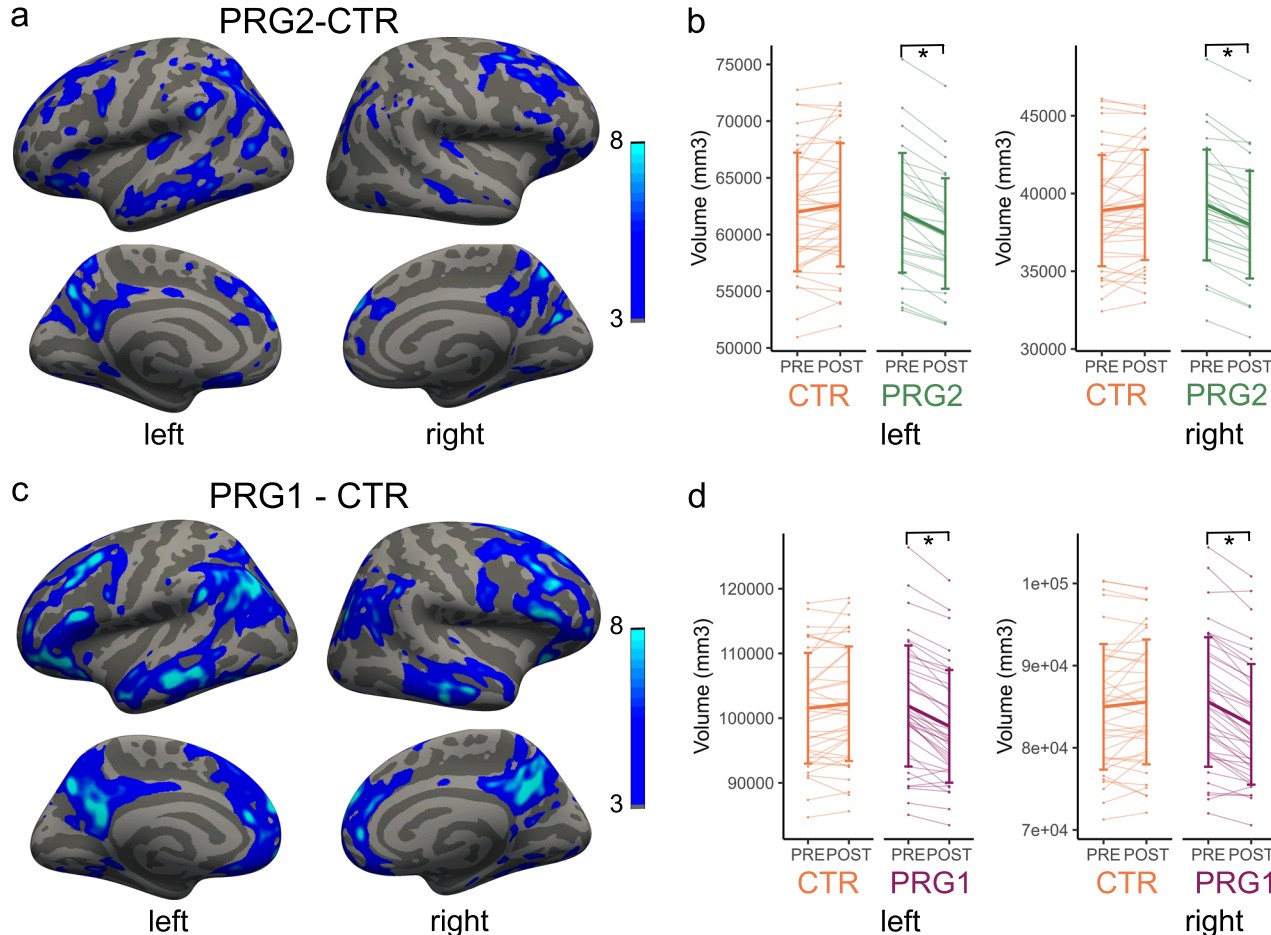

**Fig. 1 | Vertex-wise analysis of cortical volume change between multiparous, primiparous and control women.** Vertex-wise analyses showing brain areas in which the cortical volume decreased from pre-pregnancy (PRE) to early postpartum (POST) in second-time mothers (PRG2; $n = 30$; **a**) and in first-time mothers (PRG1; $n = 40$; **c**) versus control women (CTR; $n = 40$). Values represent −log(FDR-

corrected $p$ value), thresholded at 3 (being $p = 0.001$). Extracted total volumes of all the significant vertices are shown in **b** (PRG2-CTR) and **d** (PRG1-CTR), with *$p < 0.001$ for two-sided paired $t$-tests (PRG2-CTR) or two-sided Wilcoxon signed rank tests (PRG1-CTR) within groups. Source data are provided as a Source Data file.

no significant increases in cortical volume across pregnancy compared to control women. Similar results were found when investigating cortical thickness (Supplementary Fig. 1a) and surface area (Supplementary Fig. 1b), showing decreases in cortical thickness and surface area across a first and second pregnancy when compared to control women.

When directly comparing cortical volume changes across a first and second pregnancy using a permutation-based correction for multiple comparisons, we found several areas of significantly different volumetric change between PRG1 and PRG2 (Supplementary Fig. 2), although it should be noted that these effects do not surface when applying an FDR correction as in the comparisons with CTR women. Areas with a larger decrease in volume in PRG1 were mainly localized in the default mode network, whereas areas with a larger decrease in volume in PRG2 were localized in the dorsal and ventral attention network and the sensorimotor network. Effect sizes for these differences between a first and second pregnancy were smaller than the very large effect size of a pregnancy itself, but the contrast for PRG1 vs. PRG2 still showed a moderate to high effect size (Cohen's $D > 0.5$; Supplementary Table 1). A visual inspection of the results also shows a divergence in both the extent and localization of brain changes between these groups (Fig. 1a, c).

This divergence between brain changes across a first and second pregnancy was supported by a multivariate classification approach. We performed a multivariate pattern classification analysis in PRonTo[24,25]

(v3.0) on the gray matter volume difference maps resulting from the longitudinal symmetric diffeomorphic modeling pipeline implemented in SPM12[26]. This analysis demonstrated that, based on the gray matter volumetric changes in the brain, 80% of women could be correctly classified as becoming primiparous or multiparous in between the PRE and POST session (PRG1 vs. PRG2; $p = 0.0001$ after 10,000 permutations; Fig. 2a). When classifying women getting pregnant and women not getting pregnant in between these sessions, the accuracy is even higher (PRG2 vs. CTR: 87%, $p = 0.0001$, Fig. 2b; PRG1 vs. CTR: 94%; $p = 0.0001$, Fig. 2c). Using k-folds cross-validation with ten folds to reduce the risk of overfitting, classification accuracy dropped slightly, but was still significant for all three combinations of groups (PRG1 vs. PRG2: accuracy = 70%, $p = 0.009$; PRG2 vs. CTR: accuracy = 87%, $p = 0.0001$; PRG1 vs. CTR: accuracy = 91%, $p = 0.0001$). Adding age as a regressor to the models involving groups with age differences rendered highly similar results, both with leave-one-out cross-validation (PRG1 vs. PRG2: accuracy = 81.43%, $p < 0.0001$; PRG2 vs. CTR: accuracy = 84.29%, $p < 0.0001$) and with ten-folds cross-validation (PRG1 vs. PRG2: accuracy = 74.29%, $p = 0.0008$; PRG2 vs. CTR: accuracy = 82.86%, $p < 0.0001$.)

In the affected regions, brain volumes slightly increased in the late postpartum period (POST1) but had not returned to pre-pregnancy levels in both multiparous and primiparous women (Supplementary Fig. 3 and Supplementary Table 2). Correlation analyses between the volumetric change across pregnancy and age of the first child at the

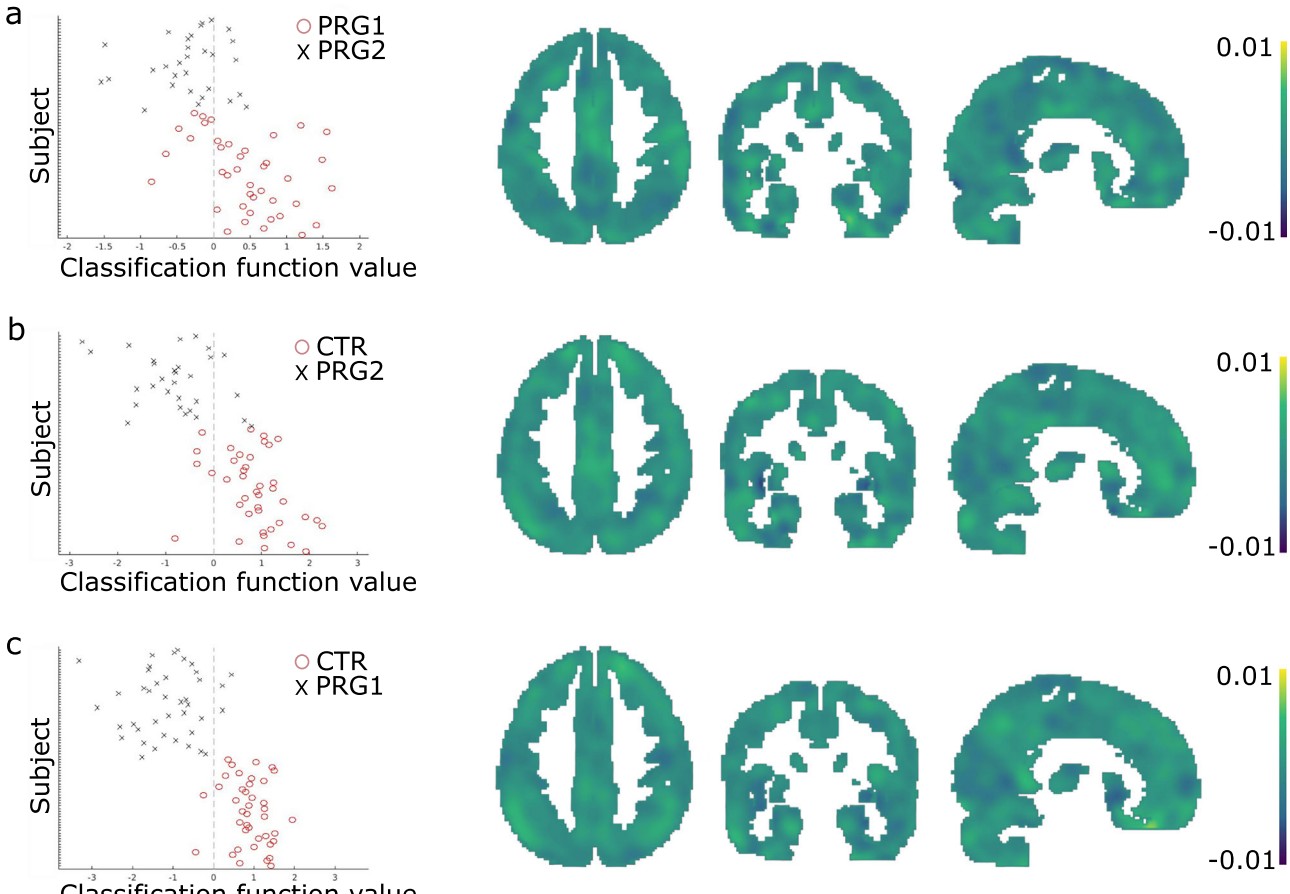

**Fig. 2 | Classification of groups based on volumetric brain difference maps.** Multivariate pattern classification analysis of women becoming pregnant for the first ($n = 40$) or second time ($n = 30$) based on brain volume difference maps (**a**). On the left side the scatterplot of the classification results (balanced accuracy 80%), showing function values; dashed line is the function value cut-off between groups, using a leave-one-out cross validation with permutation testing (10,000

permutations; $p = 0.0001$) to correct for multiple comparisons. On the right side is the weight map for the classifier, showing the relative contribution of each voxel. Similarly, the multivariate classification between control women ($n = 40$) and women getting pregnant for the second time (**b**; balanced accuracy 87%; 10,000 permutations; $p = 0.0001$) and for the first time (**c**; balanced accuracy 94%; 10,000 permutations; $p = 0.0001$). Source data are provided as a Source Data file.

PRE session in the PRG2 group suggested that time since birth of the first child does not influence the amount of volumetric brain changes across a second pregnancy (Supplementary Fig. 4). We also tested whether there were baseline differences in cortical volumes between groups, showing no significant differences between PRG1, PRG2 and CTR women in cortical volumes at the pre-pregnancy baseline. Extracting gray matter volumes inside the area of change across a first pregnancy, suggests a partial volume reversal across the postpartum period that seems to continue after the first postpartum year (Supplementary Fig. 5).

### Associations with maternal behavior and mental health

We then assessed the relationship between the volumetric brain changes across a second or first pregnancy and maternal behavior and mental health status. When comparing PRG1 and PRG2, we did not detect any significant differences in maternal behavior and mental health status (Supplementary Table 3), although we found a trend towards more depressive feelings during pregnancy in the PRG2 group. Correlation analyses showed that maternal behavior during pregnancy and in the early postpartum period was negatively correlated with the amount of volumetric change across pregnancy, except for the postpartum bonding questionnaire (PBQ) which showed a positive correlation due to its reversed scoring where higher values represent lower maternal behavior (Supplementary Fig. 6), with stronger changes in brain structure thus being associated with higher levels of maternal behavior on the employed measures. These correlations between brain volumetric changes and maternal behavior seem to be more widespread across a first compared to a second pregnancy (Supplementary Fig. 6 and Supplementary Table 4). While the volumetric changes across a first pregnancy were associated with maternal-fetal attachment as measured with the maternal antenatal attachment score (MAAS) and prenatal attachment inventory (PAI), this was not evident in second-time mothers. Associations between volumetric changes and measures of postpartum mother-infant attachment and impairments in the mother-infant relationship as measured with the maternal postnatal attachment score (MPAS) and PBQ were observed in both PRG1 and PRG2, but these were more widespread in PRG1. Nesting behavior was similarly associated with mothers' brain changes across a first or a second pregnancy.

Regarding maternal mental health, we found associations between the observed brain changes with peripartum depression, measured with the Edinburgh Postnatal Depression Score (EPDS) and psychological distress measured with the K10 in both PRG groups, with less pronounced brain changes linked to more depressive complaints for most significant vertices. However, more widespread correlations were observed between volumetric change and depression and psychological distress in the PRG2 group during pregnancy, but in the PRG1 group in the early postpartum period (Supplementary Fig. 7 and Supplementary Table 5).

### Network organization

To further investigate the regional distribution of changes in gray matter structure occurring across a first and second pregnancy, we assessed the differences and similarities between the changes in these groups in relation to the brain's functional networks using a cluster-wise approach (for cluster-wise results, see Supplementary Tables 6−9). Therefore, a combination of clusters resulting from the two significant contrasts (PRG2 vs. CTR and PRG1 vs. CTR) were applied on the volumetric GLM results described above. This resulted in three different areas of interest: Areas that are affected in both first and second pregnancies (overlap), areas that are only affected during a first pregnancy (only CTR-PRG1) and areas that are only affected during a second pregnancy (only CTR-PRG2) (Fig. 3a).

To acquire more information on the localization of these areas across the brain, we then determined their intersection with the seven resting-state networks of Yeo[27] (Fig. 3b). These analyses showed that the overlapping areas in first and second pregnancies were mostly located in the default mode network (DMN), followed by the frontoparietal and ventral attention network. For these networks, the intersection between the overlapping areas and the networks was larger than expected based on a random distribution across the gray matter in the brain (Supplementary Table 10 and Fig. 3c). Interestingly, the additional areas that were specifically affected only across a first pregnancy were localized in those same networks (Fig. 3c and Supplementary Table 11). In comparison, the areas affected in a second pregnancy only were not localized in the default mode network but instead were mainly located in the somatomotor and dorsal attention network (Supplementary Table 12 and Fig. 3c, d).

To study functional network organization, we also analyzed resting-state functional MRI data, but we did not find significant changes in within-network coherence between a first or second pregnancy in any of the identified resting-state networks. Subsequent analyses using a default mode network (DMN) region that represents the area of significant change in functional coherence across a first pregnancy[2], showed a significant group*session interaction effect between PRG1 and PRG2 in the DMN (MNI coordinates (x y z) = 12 −84 15, $T = 2.85$, $p = 0.006$ FWE-corrected; Supplementary Fig. 8), characterized by an increase in DMN network coherence only across a first pregnancy. Results for between-network connectivity are presented in the supplement (Supplementary Tables 13–15).

### White matter tract organization

We also investigated the effects of a first and second pregnancy on the organization of white matter tracts in the brain, measured with diffusion-weighted MRI. Based on these diffusion-weighted images, we extracted mean fractional anisotropy (FA) and mean diffusivity (MD in $10^{-5}$ mm$^2$/s) in 11 large white matter tracts across the brain (Fig. 4a) and compared PRE- and POST-pregnancy measurements between PRG2 and CTR subjects, PRG1 and CTR subjects, and between PRG2 and PRG1 subjects.

General linear models showed significant group*session interaction effects in the MD of the right corticospinal tract (CST) between PRG2 and CTR ($F(68) = 6.28$, $p = 0.03$, $\eta^2 = 0.15$, 95% CI [0.02, 0.29]) and in the FA of the left superior longitudinal fasciculus (temporal part; SLFT) between PRG1 and CTR ($F(78) = 7.86$, $p = 0.02$, $\eta^2 = 0.16$, 95% CI [0.03, 0.30]) (Fig. 4b and Supplementary Tables 16 and 17). These effects were also visible when directly comparing subjects undergoing a first or second pregnancy, although the FA effect in the left SLFT did not survive the FDR correction for multiple testing (MD CST: $F(68) = 7.44$, $p = 0.02$, $\eta^2 = 0.18$, 95% CI [0.03, 0.32]; FA SLFT: $F(68) = 4.09$, $p = 0.09$, $\eta^2 = 0.10$, 95% CI [0.001, 0.23]; Supplementary Tables 16 and 17). Subsequent paired $t$-tests/Wilcoxon signed rank tests revealed that these effects were driven by decreases in the MD of the right CST in the PRG2 group ($V = 386.5$, $p = 0.02$, $r = 0.58$, 95% CI [0.28, 0.78]) and decreases in the FA of the left SLFT in the PRG1 group ($t(39) = 3.74$, $p = 0.01$, $d = 0.59$, 95% CI [0.26, 0.93]). We did not observe any differences in the organization of white matter tracts at the PRE session between all three groups.

To examine if these changes were maintained across the postpartum period, we also analyzed the MD in the subset of PRG2 participants of which we had complete PRE, POST and POST1 measurements ($n = 14$). These analyses showed that the MD in the right CST was still decreased 1 year postpartum compared to the pre-pregnancy baseline ($t(13) = 4.39$; $p = 0.02$, $d = 1.17$, 95% CI [0.49, 1.85]), although in this subgroup of women there was no significant difference between the pre-pregnancy baseline and the early postpartum

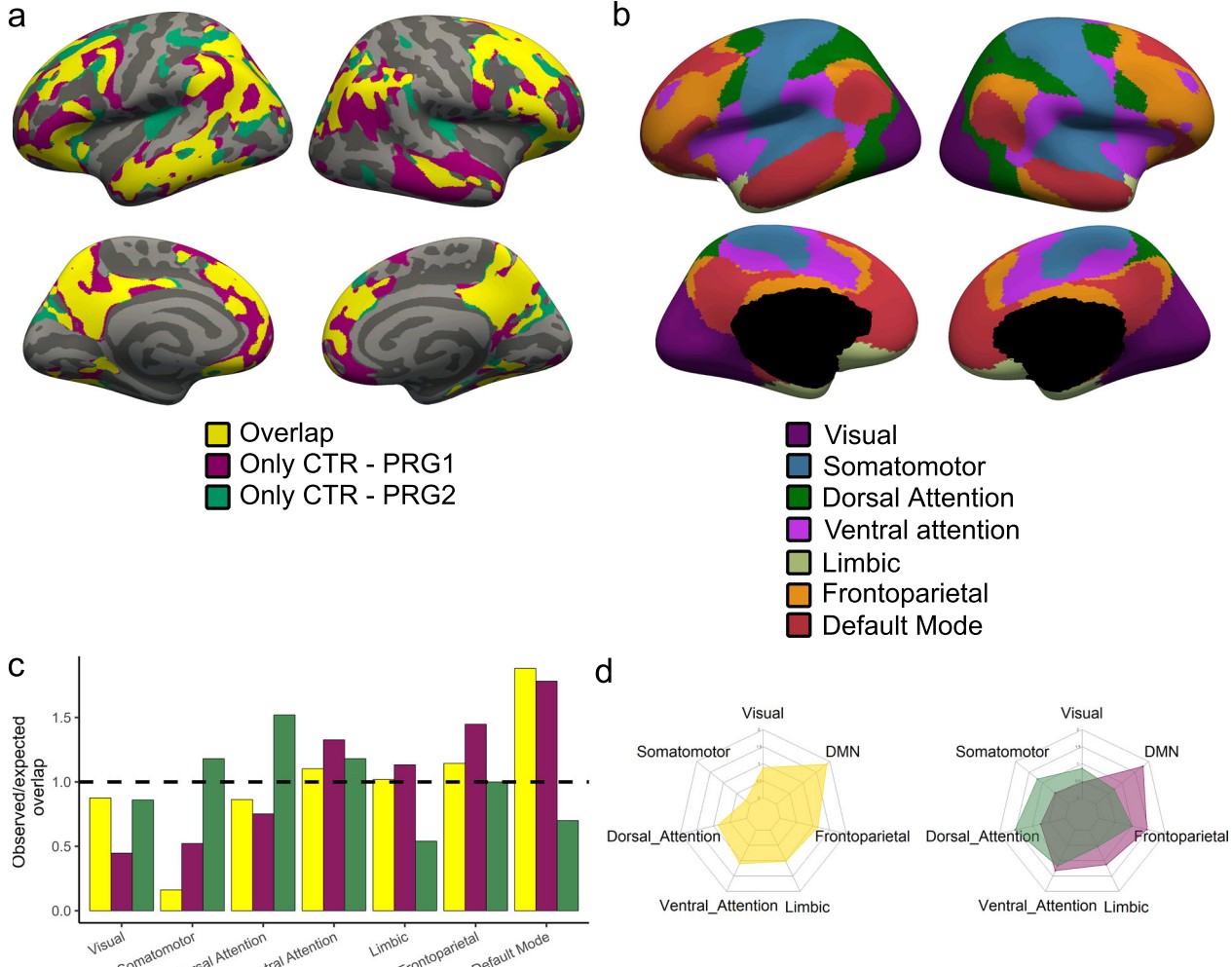

**Fig. 3 | Overlapping and differing affected areas in the brain across a first and second pregnancy.** Combining the cluster-corrected vertex-wise analyses of the multiparous (PRG2; $n = 30$) vs. control women (CTR; $n = 40$) and primiparous (PRG1; $n = 40$) vs. CTR resulted in three areas of interest: areas affected in both first and second pregnancies (Overlap; yellow), areas only affected in a first pregnancy (Only CTR-PRG1, magenta) and areas only affected in a second pregnancy (Only CTRL-PRG2; green) (**a**). We localized these areas according to the seven resting-state networks of Yeo[27] (**b**). After calculating the expected overlap based on a random distribution across the gray matter in the brain, we calculated the observed/expected ratio for each area for each functional network (**c**). The horizontal line represents an observed/expected overlap of 1, meaning that the observed intersection of the area is similar as when there would be a random distribution across the brain. Spider plots representing the distribution across the seven Yeo networks for the three areas of interest (overlap: yellow; Only CTR-PRG1: magenta; Only CTR-PRG2: green) (**d**). Source data are provided as a Source Data file.

period after correction for multiple testing ($t(13) = 2.45$, $p = 0.18$, $d = 0.65$, 95% CI [0.08, 1.23]).

### Neural metabolites

To assess the influence of a second pregnancy on neural metabolite concentrations, we measured the concentration of five major metabolites, in the precuneus/posterior cingulate cortex VOI, an area showing strong volume reductions across a first pregnancy[1]: total NAA (tNAA: N-acetylaspartate including contributions from N-acetylaspartylglutamate), total creatine (tCr: creatine and phosphocreatine), total choline (tCho: phosphorylcholine and glyceropho-sphorylcholine), Glu (Glutamate), and Ins (myo-Inositol) (Fig. 5a). At the baseline measurement, there were no significant differences in metabolite concentration between the three groups.

We compared the PRE and POST measurement between multiparous and control women, and between multiparous and primiparous women (Fig. 5b). When examining the changes in metabolite concentration between PRG2 and CTR with general linear models, we observed a group*session interaction effect for tCr only, although this effect did not survive the correction for multiple testing ($F(62) = 4.24$, $p = 0.08$, $\beta$std = 0.31, 95% CI [0.01, 0.61]) (Fig. 5b). Subsequent one sample-tests showed that this effect was driven by significant increases in tCr across a second pregnancy ($t(26) = -5.07$, $p < 0.0001$, $d = -0.98$, 95% CI [−0.85, −0.36]). Results for the other metabolites can be found in Supplementary Table 18. Additionally, directly comparing PRG2 and PRG1 did not show any significant differences in metabolite concentration changes across a first and second pregnancy (Supplementary Table 19).

## Discussion

Although we are starting to unravel the drastic neuroplasticity associated with a first pregnancy, it is currently unknown how a woman's brain is transformed by undergoing another pregnancy. Therefore, we performed a prospective pre-conception cohort study involving 110 women and acquired multimodal imaging data, including high-resolution anatomical MRI, resting-state functional MRI, diffusion-weighted MRI and magnetic resonance spectroscopy. This allowed us to investigate the effects of a second pregnancy on gray matter

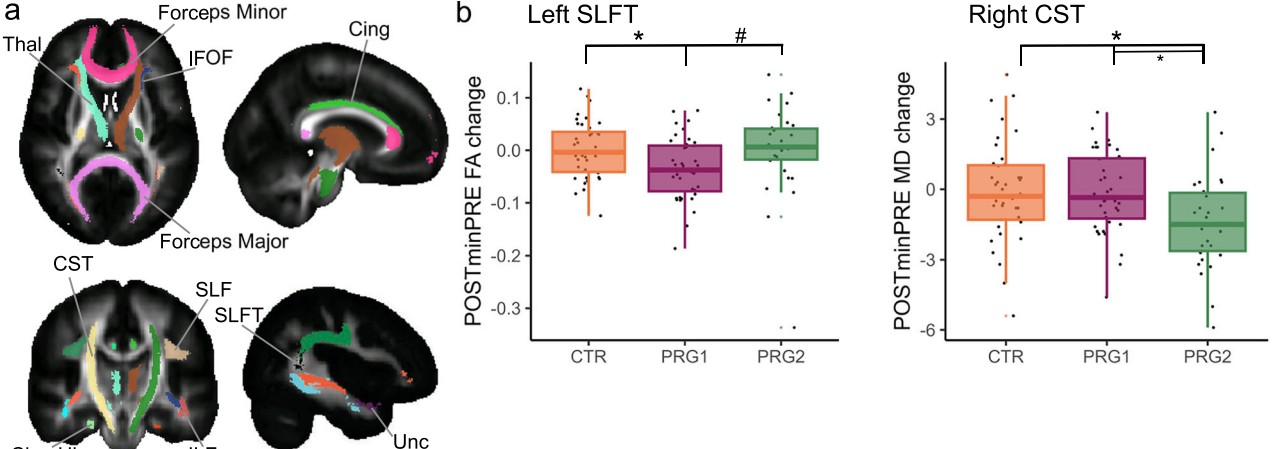

**Fig. 4 | Changes in white matter organization across a first and second pregnancy.** We investigated the white matter organization of eleven large white matter tracts in the brain (**a**). Extracted fractional anisotropy (FA) and mean diffusivity (MD; in $10^{-5}$ mm²/s) in white matter tracts that showed significant changes from pre-pregnancy (PRE) to the early postpartum period (POST) across first (PRG1; $n = 40$) and/or second (PRG2; $n = 30$) pregnancies and control women (CTR; $n = 40$) (**b**). *$p < 0.05$ for two-sided paired $t$-tests after FDR correction for multiple testing, #$p < 0.1$ for two-sided paired $t$-tests after FDR correction for multiple testing. The CTR-PRG1 comparison in the left SLFT revealed $p = 0.03$, the CTR-PRG2 comparison in the right CST $p = 0.03$ and the PRG1-PRG2 comparison in the right CST $p = 0.03$, all after FDR correction. Thal thalamic radiation, CST Corticospinal Tract, IFOF Inferior Fronto-Occipital Fasciculus, Cing Cingulum bundle, Cing Hipp Cingulum bundle (Hippocampal part), SLF Superior Longitudinal Fasciculus, SLFT Superior Longitudinal Fasciculus (Temporal part), ILF Inferior longitudinal fasciculus, Unc Uncinate Fasciculus. Boxplots show the median and interquartile range (IQR); whiskers extend to 1.5x IQR. Individual points (jittered) are overlaid for visualization and include all values. Source data are provided as a Source Data file.

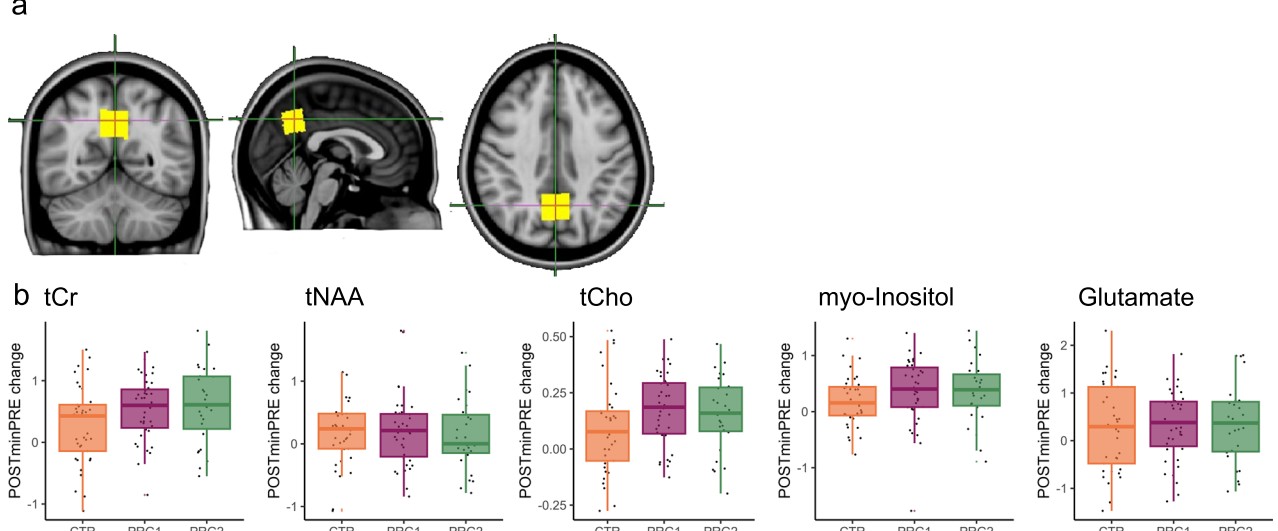

**Fig. 5 | Changes in metabolite concentrations across a first and second pregnancy.** Neural metabolites were measured in a voxel placed in the precuneus/posterior cingulate (**a**). Extracted changes in tCr, tNAA, tCho, myo-Inositol and Glutamate concentration (mM) from pre-pregnancy (PRE) to the early postpartum period (POST) across a first (PRG1; $n = 39$) and second (PRG2; $n = 27$) pregnancy and in control women (CTR; $n = 37$) (**b**). tCr total creatine (creatine and phosphocreatine), tNAA total N-acetylaspartate including contributions from N-acetylaspartylglutamate, tCho total choline (phosphorylcholine and glycerophosphorylcholine). Boxplots show the median and interquartile range (IQR); whiskers extend to 1.5x IQR. Individual points (jittered) are overlaid for visualization and include all values. Source data are provided as a Source Data file.

volume, resting-state brain activity, white matter tract organization and neural metabolite concentrations in comparison to women undergoing a first pregnancy and nulliparous control women.

These results revealed that widespread reductions take place in cortical brain volume across a second pregnancy compared to control women, which are similar to the structural changes seen across a first pregnancy. Nevertheless, classification and cluster-wise analyses demonstrated that a second pregnancy differentially affects women's gray matter brain structure compared to a first pregnancy, identifying differences in the patterns of neural change and the degree to which

different networks are affected. The observed divergence in the affected networks across a first and second pregnancy was also supported by our functional MRI data showing that functional coherence in the default mode network only increases across a first pregnancy. Furthermore, changes in white matter tract organization in different neural networks were observed across a first and second pregnancy. Together, these results demonstrate that a first and second pregnancy induce similar but also distinctive effects on a woman's brain.

The widespread cortical volume reductions we demonstrated across a second pregnancy resemble what we and other independent

research groups have shown as brain plasticity across a first pregnancy[1–5]. However, a multivariate pattern recognition approach showed that women could significantly be classified as having undergone a first or a second pregnancy based solely on their brain changes between sessions. Whereas a previous cross-sectional study in mothers did not show differences in brain volume between primiparous and multiparous women in the early postpartum period[28], our longitudinal approach reveals that the brain is differently affected across a first and second pregnancy. Deviations between reproduction-related neuroplasticity in primiparous and multiparous mothers have also been demonstrated in animal studies showing different cellular and molecular signatures in the early postpartum period[6–8]. Whereas multiparity has been associated with lower levels of hippocampal amyloid precursor protein (a marker of neurodegeneration)[8] and enhanced spine density in the CA1 region of the hippocampus[6], primiparity has been associated with dendritic remodeling in the CA1 and CA3 subregions of the hippocampus[6] and a decrease in cell survival in the dentate gyrus[7]. Although our MRI-based measures of neuroplasticity cannot provide information about the underlying cellular processes across a first and second pregnancy, our data show that, similar to rodents, reproductive experience also differently influences the brain of human mothers.

Subsequent analyses examining differences in brain plasticity across a first and second pregnancy reveal both distinct and overlapping neural networks involved. The overlapping areas affected across a first and second pregnancy were mostly located in the default mode network (DMN), followed by the frontoparietal and ventral attention network. We know from previous research that these introspective and higher-order cognitive networks are strongly affected during a first pregnancy, both structurally[1–3,5] and functionally[2]. Our findings suggest that these networks represent the main networks associated with pregnancy-induced brain plasticity, being it the first or a subsequent pregnancy. Interestingly, the additional areas that were specifically affected across a first pregnancy were localized in those same networks. This suggests a primary adaptation of these networks in women who become a mother for the first time, which is then further fine-tuned in a similar but more subtle way during a second pregnancy. Our resting-state functional connectivity analyses show a similar pattern. While an increase in DMN network coherence was found in the cuneus of the DMN network in a first pregnancy[2], this functional change did not occur to the same degree across a second pregnancy, pointing to a primary structural and functional adaptation of this network when becoming a mother for the first time. The increase in DMN coherence opposes the effect of healthy aging on DMN connectivity, which is characterized by a decrease in within-network DMN connectivity with increasing age[29] and associated with changes in cognitive functioning[30]. On the other hand, psychiatric disorders have shared and disorder-specific patterns of within-network DMN connectivity, characterized by increases and decreases across various diagnosis[31]. Therefore, more research is needed to fully understand the functional implications of increased DMN connectivity across a first pregnancy.

It has been well-established that the default mode network plays an important role in introspection, self-perception and social cognition[32–34], and the cuneus—the main area where we found the differential effect of a first and second pregnancy on functional network coherence—represents a core structure subserving the neural representation of the self[32]. Changes in the DMN across a first pregnancy have been interpreted as shifts in a mother's self-perception and her ability to understand her children's needs and feelings. Indeed, we have previously shown that pregnancy-related brain changes in the default mode network relate to a mother's neural, physiological and emotional reactions to her infant[1,2,35,36] and the degree to which she differentiates her fetus from herself during pregnancy[2]. Similarly, our current results show that the volumetric changes across pregnancy are related to maternal behavior. These correlations are more widespread in a first compared to a second pregnancy, suggesting that brain changes in a first pregnancy more strongly contribute to the induction of maternal behavior, whereas in a second pregnancy these changes play a smaller role as maternal behavior was already developed across the first pregnancy. Furthermore, the volumetric brain changes were also associated with peripartum depression and psychological distress in both first-time and second-time mothers, suggesting that pregnancy-induced brain changes play a role in the development of disorders of maternal mental health. Interestingly, the volumetric brain changes were more prominently associated with mental health status during a second pregnancy, but with mental health status in the postpartum period after a first pregnancy. We can speculate that his may be due to higher stress levels during a second pregnancy since the mother needs to care for another child during her pregnancy, although we did not find significant differences in stress and depression across a first and second pregnancy. More research investigating the neural substrates of maternal mental health disorders is needed to further elucidate these findings.

Similar to the DMN changes across a first pregnancy, we found a primary modulation of the frontoparietal network in first-time mothers, namely a decreasing FA in the temporal part of the superior longitudinal fasciculus, also known as part of the arcuate fasciculus, compared to control participants but not compared to women undergoing a second pregnancy. Although a matter of debate, lower FA values might indicate reduced integrity of the superior longitudinal fasciculus, as water diffusion is less directional, suggesting a less organized tract. This tract plays an important role in language processing, and enables communication of the frontoparietal regions and the temporal lobe[37]. Disruptions in white matter integrity have been associated with cognitive dysfunction in different disorders, such as an association between lower FA values in the superior longitudinal fasciculus and working memory deficits and speed of processing in Schizophrenia[38]. In accordance with the relatively pronounced cortical volume changes in first-time mothers in the frontoparietal network[2,3], these results suggest that a first pregnancy more strongly transforms this higher-order cognitive neural network compared to a second pregnancy.

In comparison, the areas affected specifically across a second pregnancy were not localized in the introspection-related default mode network and the cognitive frontoparietal network but instead were mainly located in networks involved with the responsiveness to external stimuli, goal-oriented attention and task demands, such as the somatomotor and dorsal attention networks. These changes can be speculated to prepare a woman for the increased demands associated with caring for multiple children at the same time. Indeed, a previous study showed different neural responses as an index of attention to both social and non-social visual stimuli during pregnancy in primiparous compared to multiparous women[39]. The changes in the sensorimotor network across a second pregnancy were also supported by our diffusion-weighted MRI findings. In women undergoing a second pregnancy, mean diffusivity (MD) in the right corticospinal tract reduced across a second pregnancy, compared to control participants and women pregnant of their first child. A reduction in MD in white matter indicates that water molecules are diffusing less freely in all directions within the tissue, potentially reflecting an increase in the structural integrity. The corticospinal tract is the main white matter tract conveying motor and sensory signals from and to the sensorimotor network[40]. It has been suggested that a change in MD may reflect synaptic plasticity, and may be a biomarker for microstructural changes associated with learning[41,42]. In comparison, aging is associated with increases in MD across the white matter[43], suggesting that pregnancy may oppose this aging effect. However, further research is needed to elucidate the functional implications of these changes in brain structure in second-time mothers.

When analyzing the magnetic resonance spectroscopy data measuring neural metabolite concentrations in the precuneus/posterior cingulate voxel, we did not find robust changes that survived a correction for multiple testing across a second pregnancy. This is in line with our previous study that found indications for changes in myo-inositol, total creatine and total choline concentrations across a first pregnancy, but these also did not survive the multiple testing correction[2]. These results suggest that there is no clear effect across a first or second pregnancy on metabolite concentrations in the precuneus/posterior cingulate cortex.

Comparisons of the baseline MRI data revealed no significant group differences in any of our presented neural measures of vertex-wise brain volumes, resting-state functional connectivity, white matter tract organization and neural metabolites. This is in line with our findings of changed temporal coherence in the DMN, which reverts to baseline across the first year postpartum[2]. Previous findings showing changes in white matter organization also indicate that these effects are transient[4]. However, the changes in gray matter structure seem to be long-lasting. Although a partial reversal to pre-pregnancy baseline has been shown across the postpartum period[1-4], structural alterations were still evident 2[1] and even 6 years[44] after delivery. While the volumetric means at the PRE sess are lower in PRG2 women in brain areas that undergo reductions across a first pregnancy, these are not statistically significantly different. This lack of observed differences in gray matter structure in the pre-pregnancy session between the PRG2 and other groups in our study could be due to the reduced power and sensitivity of cross-sectional analyses, especially because the variation in gray matter volumes at the PRE timepoint is very large. Additionally, the postpartum recovery process is highly dynamic[45], with volumetric increases shown across different time points postpartum[46-49], but also volumetric decreases from 1 to 2 years postpartum[50]. The large range of postpartum time since the first pregnancy in our second-pregnancy group may have masked baseline differences between the PRG2 and other groups in our current study. Future studies examining brain changes across a first and second pregnancy within the same women may provide a deeper insight into the recovery of brain changes in between subsequent pregnancies and the shared and distinct neural effects of successive pregnancies.

Although our manuscript focusses on relatively short-term effects of parity on the brain, studies in late life showing associations between parity and brain structure in middle-aged women suggest that traces of pregnancy-related neural changes may be present throughout the lifespan. Middle-aged women who had undergone multiple pregnancies showed younger-looking brains compared to primiparous and nulliparous women[14]. Additionally, cortical thickness and functional connectivity was related to parity in elderly women[15,16], and reproductive experience may influence the risk for and effects of different diseases in later life, like Alzheimer's Disease and stroke[18,23]. Similarly, in middle-aged rodents, long-term effects of parity on the immune system and reduced brain aging associated with reproductive experience have been shown, characterized by more neurogenesis, higher levels of brain derived neurotrophic factor and more synaptic proteins in the hippocampus[9,10,51]. Additionally, the response to ovarian hormones at middle age seems to be related to parity in rodents[11,12]. The differential neural effects of a first and second pregnancy we showed may contribute to these neural effects of parity in late life.

Various limitations of our study need to be considered. First, due to local ethical constraints, we were not allowed by the ethical committee to acquire MRI scans during pregnancy. Therefore, the exact timing of the pregnancy-induced changes cannot be concluded from our analyses. Nevertheless, previous studies examining changes during pregnancy have consistently replicated the changes demonstrated in a pre-post pregnancy design[3-5]. This strong consistency of pregnancy-induced volumetric decreases measured at different time points across pregnancy, characterized by an inverted U-shape from pre to post-pregnancy with the lowest volumes during the third trimester of pregnancy[3-5], supports the notion that the changes we observed are induced by pregnancy.

Although we demonstrate clear effects on MRI-based measures of cortical volume and white matter tracts across pregnancy, our study cannot reveal any information about the cellular processes that are underlying these changes. Gray matter volume reductions may reflect neurodegeneration, although our previous research has demonstrated a high similarity between morphometric change in pregnancy and adolescence, suggesting that pregnancy-induced brain plasticity may rather reflect a fine-tuning process[52,53]. Additionally, the cellular processes underlying changes in diffusion-based metrics in white matter tracts are a matter of debate[54], and changes in FA or MD could reflect different processes underlying neuroplasticity and learning, such as astrocyte swelling, dendritic spine changes, angiogenesis or synaptic changes[41,55,56]. Nevertheless, decreases in MD have also been shown during adolescence[57,58], further supporting the notion that pregnancy-induced neuroplasticity resembles brain plasticity across adolescence. Results regarding white matter fractional anisotropy or quantitative anisotropy changes in white matter across pregnancy have not been conclusive, with our previous study showing no changes from pre-pregnancy to the early postpartum period[2] and another study showing increasing quantitative anisotropy during pregnancy, returning to baseline postpartum in a single women undergoing her first pregnancy[4]. Different timings of research sessions and different analysis strategies may explain these discrepancies, and more research is needed to fully elucidate the effect of pregnancy on white matter tracts.

Although our study includes a relatively large group of women, especially given the complicated nature and logistics of our longitudinal study design, the group sizes are still limited for classification analyses. To control for the risk of overfitting in small sample sizes, we performed a leave one subject out cross-validation scheme, keeping the training dataset as large as possible to train our classifier. Since it has been shown that the leave-one-out cross-validation can potentially lead to over-optimistic model performance estimations, we repeated our analyses with a k-fold cross-validation scheme with 10 folds, which resulted in a slightly lower accuracy but still significant classification between PRG1 and PRG2. Nevertheless, future research with larger group sizes would be beneficial to confirm our findings.

Because of time constraints during data acquisition, we were able to collect only 5 min of resting-state fMRI data. These data are valuable to compare to our previously published results about resting-state functional connectivity changes in the DMN across a first pregnancy[2] that was acquired with the same MRI protocol, but we acknowledge that according to newer standards these data may be limited and should be interpreted with caution, and are therefore presented in the supplement. Future studies acquiring more resting-state fMRI volumes are needed to confirm changes in resting-state functional connectivity across pregnancy.

Lastly, although we tried to match our groups based on demographic variables, there was a significant difference in age between second-time mothers and first-time mothers and control women. Since age is also known to influence cortical volumes[59], resting-state functional connectivity[60,61] and diffusion-based structural connectivity[61], we corrected for age in all analyses. Since these aging effects on functional and structural connectivity are in the opposite direction compared to the pregnancy-induced effects we have found, we do not expect age differences to solely underlie our results. Additionally, there may be other potential confounding factors related to pregnancy that influence the observed changes across a second pregnancy, such as age of first pregnancy, breastfeeding, type of delivery and other contributing factors, such as sleep disturbances, stress or social

support. Although our previous findings suggest that several of these factors including sleep, stress, type of delivery and breastfeeding do not strongly contribute to pregnancy-induced brain changes[2], it is likely that such factors are associated with relatively subtle effects that only surface in large samples. Studies involving a larger group of women becoming mothers may be able to acquire more insights into the influence of such factors on pregnancy-induced brain changes.

In conclusion, we have demonstrated widespread volumetric decreases across a second pregnancy, which are highly similar to the effects we and others have shown across a first pregnancy[1–5]. Despite these similarities, women could be classified as having undergone a first or second pregnancy solely based on the changes in their brain structure, indicating that subsequent pregnancies are associated with distinct neural transformations. Both a first and second pregnancy particularly strongly impacted the introspective default mode network and the frontoparietal network and one of its major white matter trats, the superior longitudinal fasciculus. However, these changes were more prominent in a first pregnancy, suggesting a primary adaptation of this network in women who become mothers for the first time that is further fine-tuned during a second pregnancy. Accordingly, the changes in the temporal coherence in the default mode network observed in a first pregnancy were not found in second-time mothers. On the other hand, second-time mothers exhibited stronger structural alterations in the dorsal attention and somatomotor networks including the corticospinal tract, suggesting that a second pregnancy entails an enhanced plasticity within these externally-oriented networks. Correlation results revealed associations of volumetric brain changes across both a first and a second pregnancy with mother-infant attachment, but these were more widespread in first-time mothers. Furthermore, the structural brain changes across both a first and a second pregnancy were associated with maternal mental health, but these were more prominently associated with postpartum depression and psychological distress in the postpartum period in first-time mothers and with depression and psychological distress during pregnancy in second-time mothers. These findings demonstrate that the human brain is altered across a second pregnancy, involving changes in gray matter structure, white matter tracts and resting-state brain activity, and show that both a first and second pregnancy confer a unique mark on a woman's brain.

## Methods

### Study set-up and sample

This research was evaluated and approved by the Ethics Review Board of the Leiden University Medical Center and complies with all relevant ethical regulations. All participants signed the informed consent forms before any study-related measurement and received monetary compensation. We used a prospective pre-conception cohort study, in which we followed women with the intention to become pregnant of their second child in the following year (PRG2: $n = 30$). Only women (self-reported) were included in this study. We started with a pre-conception research session (PRE session), followed by a session during the third trimester of pregnancy, a session in the early postpartum period (80.32 ± 27.72 days from parturition; POST session) and a session in the late postpartum period (408.67 ± 38.75 days from parturition; POST1 session). Because of the COVID-19 pandemic, only a subset of women could participate in the POST1 session ($n = 14$). Due to this low sample size, we just included the POST1 timepoint in analyses where we found a significant pregnancy effect from PRE to POST in the PRG2 group, to study the long-term effects of pregnancy. The pregnancy session only included questionnaires, hormone sampling and cognitive testing, since MRI scanning was not allowed during pregnancy by the ethical committee. Additionally, this study included a primiparous group, including women who became pregnant of their first child during the course of the study (PRG1: $n = 40$, for the POST1 session $n = 28$) and nulliparous control women (CTR: $n = 40$), who underwent a similar study set-up (see Fig. 6)[2]. All groups were scanned in the same time period, and individuals of different groups were scanned intertwined. For more information about the sample included, see Supplementary Table 20.

The PRG2 group was significantly older (32.03 ± 2.33 years) than the PRG1 (29.35 ± 3.51 years; $p = 0.0027$) and the CTR group (29.33 ± 3.57 years; $p = 0.0025$), so age has been used as a covariate in all analyses. There were no significant differences in level of education, time between PRE and POST sessions, time between birth and POST session and time between birth and POST1 session (Supplementary Table 20).

### MRI acquisition

All MRI scans were acquired on a Philips 3 T MRI system.

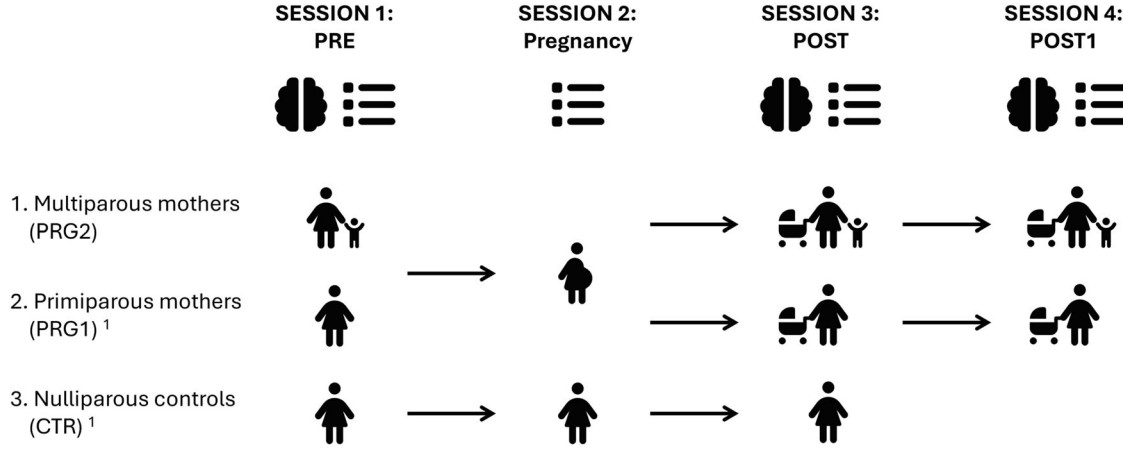

**Fig. 6 | Study design.** We included second time mothers (multiparous; PRG2), first-time mothers (primiparous; PRG1) and control women (CTR) not getting pregnant in between the scans (CTR). We acquired structural T1-weighted, resting-state functional, diffusion-weighted MRI scans and a magnetic resonance spectroscopy (MRS) scan before pregnancy (Session 1: PRE), in the early postpartum period (Session 3; POST) and the late postpartum period (Session 4; POST1). The pregnancy session only included questionnaires, hormone sampling and cognitive testing, since MRI scanning was not allowed during pregnancy by the ethical committee. Icons in this figure were obtained from Font Awesome (https://fontawesome.com) and are used under the Creative Commons Attribution 4.0 International License (CC BY 4.0; https://creativecommons.org/licenses/by/4.0/). The icons were not adapted. Source data are provided as a Source Data file[1]. Hoekzema et al. (2022).

**Anatomical MRI**. High-resolution 3D anatomical T1-weighted images were acquired in transverse orientation, with the following acquisition parameters: repetition time (TR) = 9.8 ms, echo time (TE) = 4.6 ms, Flip Angle = 8°, Field of View (FOV) = 178 × 224 × 168 mm, voxel size = 0.875 × 0.875 × 1 mm. All MRI scans were visually checked for quality control, and no scans had to be excluded. Additionally, we determined Freesurfer's Euler number as a measure of data quality for each T1w image at each time point (Supplementary Table 21), and did not find a significant group (PRG2, PRG1, CTR) * session (PRE, POST, POST1) interaction effect ($F(3,254) = 0.42$, $p = 0.74$). After processing in Freesurfer, all individual T1w images had Euler numbers of 2, which reflect an accurate surface reconstruction.

**Resting-state functional MRI**. Functional MRI scans were acquired in all participants during rest, while fixating at a crosshair at the screen to prevent them from falling asleep. A total of 137 T2*-weighted whole-brain echo-planar images (EPIs) and two dummy scans were acquired with the following parameters: TR = 2.2 s; TE = 30 ms, Flip Angle = 80°, FOV = 220 × 220 × 111.65 mm, voxel size = 2.75 × 2.75 mm, 37 descending slices.

**Diffusion-weighted MRI**. Two sets of transverse diffusion-weighted images (DWI) were acquired with reversed k-space encoding direction, to allow for distortion correction. For both DWI scans, the acquisition parameters were: TR = 7315 ms, TE = 69 ms, Flip Angle = 90°, FOV = 240 × 240 mm, acquisition matrix = 128 × 98, reconstruction matrix 128 × 128, 30 different diffusion directs with b-factor 1000 s/mm$^2$, 5 B0 acquisitions, SENSE factor = 3, 75 slices of 2 mm, no slice gap, no cardiac gating.

**MR spectroscopy**. Magnetic resonance spectroscopy (MRS) was performed with single-voxel point-resolved spectroscopy (PRESS) localization (TR = 2000 ms; TE = 37 ms; 128 averages, 2 dummy scans, and 16 reference scans without water suppression). Shimming was performed with an automated second order projection-based algorithm. The volumes of interest (VOIs) were positioned in two regions that were found to undergo strong changes in brain structure in our previous study[1]. One was the precuneus/posterior cingulate cortex (PCC), centered between both hemispheres (Fig. 6a), which had a volume of 8 mL (20 × 20 × 20 mm$^3$). The other VOI was positioned in the right superior temporal gyrus and had a volume of 12 mL (20 × 30 × 20 mm$^3$), but due to low quality spectra, this VOI was excluded from all analyses.

## MRI processing and statistics

**Anatomical MRI**. The anatomical MR images were processed with Freesurfer 7.2.0, using the longitudinal *recon-all* stream[62]. We performed vertex-wise analyses using cortical volume, thickness and surface area, for which the subject's cortical maps were registered to the *fsaverage* space and smoothed with a 10 mm full-width-at-half-maximum (FWHM) kernel. Cortical change maps were created for each measure, and generalized linear models (GLMs), including age as a covariate, were performed to determine differences in the vertex-wise change between each combination of groups (PRG2 vs. CTR, PRG1 vs. CTR and PRG2 vs. PRG1). These linear models were false discovery rate (FDR) corrected for multiple testing. We also determined Cohen's *D* effect sizes for the significant vertices (Supplementary Table 1). In order to allow for slightly less prominent effects to surface, we also used a more lenient approach to correct this contrast for multiple testing than the FDR-correction, using 1000 permutations and a vertex-wise threshold of $p < 0.01$, corrected for performing analyses in the left and right hemisphere. Cortical volumes were extracted from the total area as output of each GLM, and paired *t*-tests, or in case of non-normality Wilcoxon signed rank tests, were used to determine the direction of effect in the three groups. These paired *t*-tests/Wilcoxon signed rank tests were repeated in multiparous ($n = 14$) and primiparous

($n = 28$) women with complete datasets, including the late postpartum timepoint (POST1). Differences between groups at the pre pregnancy baseline were determined using linear models corrected for age.

To further investigate differences in brain changes across subsequent pregnancies, we determined whether women undergoing a first or second pregnancy could be separated based on their anatomical MRI scans. To do so, we performed linear support vector machine classification in Pronto (PRoNTo v3.0)[24,25] on the smoothed gray matter volume difference maps as output from longitudinal symmetric diffeomorphic modeling pipeline[26] in SPM12 (http://www.fil.ion.uncl.ac.uk/spm/) implemented in Matlab 7.8 (Mathworks), using a leave-one-out cross-validation. Because of our relatively small sample size for classification analyses, the leave-one-out cross validation provides the largest training dataset as possible, but at the risk of overfitting and overestimating classifier accuracy. Therefore, we repeated the analyses using a k-fold cross-validation using 10 folds, which reduces the risk of overfitting although it's better suited for large sample sizes. Details about the longitudinal pipeline in SPM12 can be found in Hoekzema et al.[2]. Permutation testing using 10,000 permutations was used to determine the significance of the classification accuracy for all three combinations of groups (PRG2 vs. PRG1, PRG2 vs. CTR and PRG1 vs. CTR).

Lastly, we performed a cluster-wise correction on the results from the above described GLMs with a vertex-wise cluster threshold of 0.001 and a cluster-wise *p*-threshold of 0.05 to determine clusters of significant volumetric differences. We combined these cluster maps of the results from PRG1-CTR and PRG2-CTR, to determine the similarity and difference in areas affected with a first or second pregnancy compared to control women. This resulted in three areas of interest: areas affected in both a first and second pregnancy (Overlap), areas affected only in a first pregnancy (Only CTR-PRG1) and areas affected only in a second pregnancy (Only CTR-PRG2). To localize these areas in the brain, we determined the intersection of those regions with the 7 resting-state networks of Yeo[27], relative to the expected volume of the intersection based on a random distribution across the gray matter in the brain.

**Resting-state functional MRI**. The resting state fMRI images were preprocessed using DPARSF (version 4.5)[63], involving slice timing correction, realignment, and co-registration of the anatomical images to the mean functional images. The transformed anatomical images were then segmented[64], and DARTEL[65] was used to transform the images to MNI space, followed by the application of a 10 mm$^3$ FWHM Gaussian kernel. To account for head motion, we applied the Friston 24-parameter model[66] and subjects with any frame-wise displacements (FD) exceeding 2 mm (for translations) or 2° (for rotations) or with a mean FD exceeding 0.2 in any of the sessions were excluded[67]. Therefore, 3 participants of the PRG2 group and 4 participants of the CTR group had to be excluded. After these exclusions, there were no significant group (PRG1, PRG2 and CTRL) * session (PRE and POST) interaction effects in the mean FD or any of the other motion parameters (maximum translations and rotations in the x, y and z-directions) (Supplementary Table 22).

We performed Group spatial independent component analyses (ICA) using the Group ICA for fMRI Toolbox in Matlab (GIFT v4.0b, http://mialab.mrn.org/software/gift), using default options, 20 components and the InfoMax algorithm. Components were selected through automated selection by spatial sorting with the components of Smith et al.[68] who defined the major networks in the resting brain using a similar ICA-based approach, using a cutoff value of $R > 0.25$. We compared within-network coherence across the obtained functional networks, by determining the functional connectivity of each voxel with all the other voxels in the network. After determination of the neural networks, we extracted the correlation between the functional networks using the Functional Network Connectivity toolbox (FNC

Toolbox version 2.3, https://trendscenter.org/software/fnc/) for Matlab with a lag-shift algorithm using default options. FNC calculates a constrained maximal lag correlation between each pair of networks by calculating Pearson's correlations and constraining the lag between the time courses[69].

To examine whether there were differences between groups in the within-network coherence change across the PRE and POST sessions, we performed generalized linear models in SPM12 and investigated the group * session interaction effect using the PRG2 and PRG1 group, and the PRG2 and CTR group. As a follow-up analysis, for the model investigating the default mode network, a region of interest representing the significant within-network coherence change across a first pregnancy[2] was applied as mask. In these analyses, we used the average functional connectivity of the voxels inside this mask only with the rest of the DMN. Results were considered significant at an FWE-corrected statistical threshold of $p < 0.05$. When significant interaction results were obtained, these were followed by paired sample $t$-tests, or in case of non-normality the non-parametric Wilcoxon signed rank test, using the PRE and POST measurement within each group. Baseline differences between groups were determined using linear models corrected for age. For the between-network correlation changes, we performed repeated-measures general linear models in Rstudio (version 23.06.1). In case of significant group * session interaction effects, we performed paired $t$-tests, or in case of non-normality the non-parametric Wilcoxon signed rank test, using the PRE and POST measurement within each group. Results were considered to be significant at a statistical threshold of $p < 0.05$, corrected for multiple testing using an FDR correction. To create images, the statistical maps were projected onto the PALS surface provided in Caret software (http://brainvis.wustl.edu/wiki/index.php/Caret). Slice overlays were created using MRIcron (http://www.mccauslandcenter.sc.edu/mricro/mricron/).

**Diffusion-weighted MRI**. Diffusion-weighted images were processed using MRtriX3[70], unless otherwise stated. We denoised the data, pre-processed the data including eddy current-induced distortion correction, motion correction and susceptibility-induced distortion correction using FSL tools eddy & top-up, and corrected the data for B1 inhomogeneity using *dwidenoise*, *dwifslpreproc* and *dwibiascorrect*, respectively. Subsequently, we fitted the diffusion-tensor and calculated the fractional anisotropy (FA) and mean diffusivity (MD) maps. Diffusion tensors were fitted in two steps using *dwi2tensor* with default settings: first, using weighted least-squares with weights based on empirical signal intensities, and second, by further iterated WLS with weights determined by the signal predictions from the previous iteration. Mean B0 maps were linearly registered using FSL (version 6.0.3) to one individual B0 map, followed by taking the mean to create a study-specific template. Afterwards, individual mean B0 maps were non-linearly registered to the study-specific template, and subsequently to the JHU-ICBM atlas template[71]. Registrations were inverted, and masks of 11 large white matter tracts of interest were registered to individual subject space. These included the Forceps Major, Forceps Minor, left and right Thalamic Radiation, left and right Corticospinal Tract, left and right Cingulum bundle (Cingulate part and Hippocampal part), Left and right Inferior Longitudinal Fasciculus, left and right inferior Fronto-Occipital Fasciculus, left and right Superior Longitudinal Fasciculus (Body and temporal part) and the left and right Uncinate Fasciculus. ROI masks in individual space were masked with a fractional anisotropy (FA) mask >0.25, to ensure inclusion of white matter, and mean FA and MD were calculated for each subject at each time point.

General linear models in Rstudio (version 23.06.1) were performed to compare changes in white matter tracts (FA and mean diffusivity (MD)) between the PRE and POST measurements across PRG2 and CTR women, PRG1 and CTR women and PRG2 and PRG1 women. In case of significant group * session interaction effects, we performed paired $t$-tests, or in case of non-normality the non-parametric Wilcoxon signed rank test, using the PRE and POST measurement within each group. Baseline differences between groups were determined using linear models corrected for age. To correct for multiple testing across all white matter tracts per measure (FA or MD), we performed FDR corrections. Lastly, for individuals with a complete PRE-POST-POST1 dataset (PRG2: $N = 14$, PRG1: $n = 28$), we performed paired $t$-tests/Wilcoxon signed rank tests between the PRE and POST1 measurements to investigate the long-term effects of pregnancy.

**MR spectroscopy**. MRS data of the PRE and POST session were available of 27 women in the PRG2 group, since the spectroscopy had to be omitted in three participants due to a lack of time. The POST1 data was excluded from the MRS analyses because data was only available of four participants in this group. Additionally, MRS data was acquired from 39 PRG1 women and 37 CTR women at the same time-points (PRE and POST). Metabolite concentrations were estimated with LCModel, (version 6.3-1 M), using a dataset containing seventeen metabolites. For this study, we considered the major metabolites tNAA (N-acetylaspartate including contributions from N-acetylaspartylglutamate), tCr (creatine and phosphocreatine), tCho (phosphorylcholine and glycerophosphorylcholine), Glu (Glutamate), Ins (myo-Inositol). These metabolites (or combinations thereof) can typically be measured with high precision (see next section about spectral quality). Concentrations were expressed using water scaling. Next, we corrected for partial volume contributions of GM, white matter and cerebrospinal fluid in the corresponding VOI, based on Sienax segmentation (FSL 5.0.10) of each subject's 3DT1 images.

Spectral quality was examined based on the full-width half maximum (FWHM), signal-to-noise ratio (SNR), and the estimated CramerRao lower bounds of each metabolite. Spectra with FHWM > 0.1 ppm (12 Hz) and/or SNR < 5 were considered poor quality. All PCC spectra in the PRG2 group had high quality, with SNR mean ± sd of 24.83 ± 1.51, FWHM 4.32 ± 0.51 Hz, and Cramer Rao lower bounds of metabolites well below 10%: tCr 2.06 ± 0.23%, tNAA 2.17 ± 0.38%, Cho 4.50 ± 0.50%, Ins 6.63 ± 0.62%, and Glu 8.00 ± 0.34%. Information about spectra quality in the PRG1 and CTR group can be found in our previous paper[2]. There were no differences in FWHM and SNR ratio across groups and time points.

The metabolite concentrations resulting from LCModel were analyzed in Rstudio (version 23.06.1), using repeated measures general linear models to assess whether the change in metabolite concentration from PRE to POST was different in PRG2 women, compared to PRG1 and CTR women. In case of significant group * session interaction effects, we performed paired $t$-tests, or in case of non-normality the non-parametric Wilcoxon signed rank test, using the PRE and POST measurement within each group. Baseline differences were determined using linear models corrected for age. These analyses were FDR corrected across the five metabolites as correction for multiple testing.

### Scales and questionnaires

**Maternal behavior**. To measure maternal behavior, we used several questionnaires during pregnancy (at the PREG session) and in the early postpartum period (at the POST session). To assess maternal-fetal attachment during pregnancy, women filled in the Prenatal Attachment Inventory (PAI[72]) and the Maternal Antenatal Attachment Scale (MAAS[73]). Additionally, to examine nesting behavior, the preparational activities during pregnancy, women filled in the Nesting Behavior Questionnaire[74]. To assess maternal-child bonding and impairments in the mother-infant relationship in the postpartum period, we asked women to fill in the Maternal Postnatal Attachment Scale (MPAS[75]) and the Postpartum Bonding Questionnaire (PBQ[76]) respectively. Questionnaires were filled in by both the PRG1 and PRG2 group, although

some subjects had missing data at a specific time point (PREG: PRG1: $n = 36$, PRG2: $n = 28$; POST: PRG1: $n = 39$, PRG2: $n = 28$).

**Maternal mental health.** We assessed mental health status during pregnancy and in the early postpartum period. To assess psychological distress, women in the PRG1 and PRG2 group filled in the K10 questionnaire[77]. Additionally, to measure signs of postpartum depression, we acquired data from the Edinburgh Postnatal Depression Scale (EPDS)[78]. For both questionnaires, data were missing from several of subjects (PREG: PRG1: $n = 36$, PRG2: $n = 28$; POST: PRG1: $n = 39$, PRG2: $n = 28$).

**Analyses.** We determined whether there was a difference in maternal behavior or maternal mental health between the PRG1 and PRG2 group using ANOVA analyses. Additionally, we performed correlation analyses in Freesurfer, to assess whether the maternal behavior or mental health status was associated with the volumetric changes across a first or second pregnancy. We applied the significant areas of change across a first or second pregnancy (PRG1 vs. CTR or PRG2 vs. CTR) as mask for these analyses, to only use the area of significant volume decrease across a first or second pregnancy. Results were corrected for multiple testing using 1000 permutations, with a vertex-wise $p$ value of 0.01, and corrected for performing the analyses across the left and right hemisphere.

### Reporting summary
Further information on research design is available in the Nature Portfolio Reporting Summary linked to this article.

## Data availability
Source data for each figure are provided with this paper and in Figshare (https://doi.org/10.6084/m9.figshare.31144273). The raw MRI data and group/demographic information generated in this study for the participants who have provided permission to share their data have been deposited in the Open Science Framework depository under the following https://doi.org/10.17605/OSF.IO/G8DNR. The deposited data are available open access. Source data are provided with this paper.

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

## Acknowledgements

We acknowledge the participants for their contribution to this study. We thank research assistants A. van der Geest, A. Glasbergen, S. Altikulac, A. van Steenbergen, R. van Dort, P. Berns en I. Langereis for coordinating the data collection for this project. This project was supported by an Innovational Research Incentives Scheme grant (Veni, 451-14-036, E.H.) by the Netherlands Organisation for Scientific Research (NWO), a NAR-SAD grant from the Brain and Behaviour Research Foundation, U.S.A. (grant number 25312, E.H.) and a grant of the Leiden University Fund / Elise Mathilde Fund (CWB 740s / 2t-03-2017 /EM) awarded to E.H. E.H. is currently supported by an ERC Starting Grant (948031, E.H.) provided by the European Research Council.

## Author contributions

M.S. analyzed data and wrote the paper, S.H. contributed to the processing and analysis of the structural data, P.J.W.P. supervised processing and analyses of the MRS data, E.C. contributed to the design and interpretation, E.H. designed the study, analyzed data, and contributed to the manuscript and interpretation. All authors evaluated the manuscript.

## Competing interests

The authors declare no competing interests.
