## [Transparent Peer Review file · Nature Communications]

The effects of a second pregnancy on women's brain structure and function

Corresponding Author: Dr Eline Hoekzema

Version 0:

Reviewer comments:

Reviewer #1

(Remarks to the Author)

This paper examined women's brain changes after a second pregnancy, an important research question given evidence that first pregnancy appears to change the maternal brain. Thirty women undergoing a second pregnancy, 40 women undergoing a first pregnancy, and 40 control women were compared. There was overlap in terms of brain change across the first and second perinatal period, but there were also some differences, with first time mothers showing stronger default mode network changes and second time mothers showing more pronounced changes in somatomotor and dorsal attention network regions. There were also differences in resting state connectivity and white matter connectivity.

This is a very impressive study that makes a significant contribution to the research literature. I believe that the paper warrants publication, and just have a few questions or clarifications.

First, one weakness is the lack of measures of parent-child bonding or any other individual differences measures (maternal mental health, parenting stress, amount of time spent with baby, etc). Without these measures, it is unclear whether differences between the first and second pregnancy are driven by hormones or other factors, such as parenting experience and feelings about the baby. If mothers are more bonded to their first child, do they show more or less neural change with the second pregnancy? If they feel stronger antenatal attachment during pregnancy, or stronger postnatal attachment after birth, does that explain larger magnitude of brain change? Even if there are no significant associations between self-report measures and brain changes, it would still be valuable to report null results because it would aid interpretation.

Was birth spacing (the length of time between the first and second pregnancies) considered in analysis? This would be interesting to measure as a continuous variable to see if brain change is greater in second-time mothers when there is more or less time elapsed after their first birth.

The paper would also benefit for more clear explanation and interpretation of what the results mean. For example, the white matter findings are interesting but not explained clearly for readers who are not familiar with metrics such as fractional anisotropy and mean diffusivity. Do the results suggest that white matter is increasing or decreasing in volume or integrity? What does this mean in terms of brain health? Do these changes appear adaptive? In terms of the resting state findings, what are the implications of default mode network coherence? Is DMN coherence associated with stress, mental health, or empathy? Drawing on citations from the human neuroscience literature beyond maternal brain studies would help to contextualize these findings.

I would also like to see more interpretation of some of the apparent contradictions in these findings. For example, the authors suggest that some brain changes (in the default mode, for example) are of greatest magnitude across the first pregnancy and then are fine-tuned more subtly during the second pregnancy. But if this is the case, and the brain has already done much of its restructuring in the first pregnancy, wouldn't we see baseline brain differences between first and second time mothers? Instead, it seems like everyone is starting from the same place. How do we reconcile that with the finding that the most profound brain changes seem to occur during the first pregnancy? If the brain reverts back to baseline, then presumably it would need to change in the same way to facilitate the next pregnancy. In a perfect world we could see a longitudinal study that follows the same mothers across a first and then a second pregnancy, but of course that study would be extremely challenging to conduct.

Reviewer #2

(Remarks to the Author)

This paper explores the effects of a second pregnancy on brain structure and function using multimodal MRI data. The authors report that a second pregnancy conferred similar though less pronounced changes in the default mode and frontoparietal network, yet stronger changes in the dorsal attentional network and somatomotor network compared to a first pregnancy. The current literature lacks a detailed characterization of how first versus second pregnancies influence brain health, and this paper addresses this gap nicely. This is a high-quality and ambitious study that tackles a critical yet underexplored question. The prospective pre-conception design is a major strength and adds valuable insight. Overall, this paper would make a substantial contribution to the literature on maternal brain plasticity. I have several comments to help temper interpretations of the findings and to acknowledge potential limitations.

The introduction is strong and clearly written. However, it is somewhat vague in its description of the effects of multiple pregnancies in both animal and human studies. I recognize there are limited papers examining the effects of previous pregnancies on brain health, but the authors should consider citing additional recent work and outlining the more specific differences between differing levels of parity. In addition, several human studies have assessed parity as a linear/continuous variable, offering relevant insight into how additional pregnancies may shape brain health.

Please provide a clearer justification for why the specific neural metabolites were selected for analyses. This would help contextualize the results and clarify their relevance to the broader literature on pregnancy-related brain changes.

Many studies that have examined the effects of parity on brain health have focused on short-term outcomes. It would strengthen the discussion to explicitly distinguish between short- and long-term effects of pregnancy on the brain. The authors briefly mention longer-term signatures of pregnancy on brain plasticity in humans. However, it would be valuable to integrate evidence from rodent models describing how previous pregnancies influence brain plasticity at middle age to provide a more comprehensive perspective.

The reported 80% classification accuracy between PRG1 and PRG2 is promising. However, given the relatively small sample size, there is a risk of overfitting. Please clarify whether any cross-validation procedures were used and discuss this limitation more explicitly. Additionally, the manuscript would benefit from a more detailed discussion of potential confounding variables in the context of pregnancy effects, including interpregnancy interval, age of first pregnancy, breastfeeding, and other relevant factors.

Reviewer #3

(Remarks to the Author)

Thank you for inviting me to review the manuscript entitled “The effects of a second pregnancy on women’s brain structure and function” by Straathof and colleagues. This study examines a remarkably rare and valuable cohort, and the questions it addresses are of clear significance to the field. Much of the prospective work to date—most notably Dr. Hoekzema’s groundbreaking studies—has focused on women experiencing their first pregnancy. While these investigations have been foundational, they overlook the fact that most women will experience multiple pregnancies in their lifetime. What happens to the brain in subsequent pregnancies? This study begins to address that critical gap.

However, I have several conceptual and methodological concerns that I would like to see addressed and clarified before I can recommend publication. I have outlined both major and minor considerations below.

1. My first concern involves the consistency in reporting throughout the Results section, as well as the conceptual meaning behind these findings.

- a. The authors find no significant differences when comparing vertex-wise GMV volume between the PRG1 and PRG2 groups, but visual inspection triggered a follow-up analysis (classification of GMV changes) that showed different takeaways. Can the authors speak to how we should reconcile this: are there true differences between PRG1 and PRG2, and/or is it based on the methodological approach?
- b. It’s not always clear when the authors are looking at POST1 in both or just the PRG2 group, and why those changes are not incorporated in all analyses. For example, it looks like POST1 sessions were explored with respect to GMV trajectories, but not incorporated into the classifier?

2. My second concern involves the nature of the resting-state data. The authors are using incredibly underpowered RS data - 137 images per session at a TR > 2 seconds – to conduct functional brain network analyses (cf. Birn Neurolmage, 2013). I believe they are too underpowered to do this reliably and this section should be preferably 1) removed entirely or 2) moved to supplement & addressed as a significant limitation. If the latter:

- a. Can the authors also provide a breakdown of head motion (e.g., FWD) for the different groups? Given the nature of the postpartum sessions in PRG1 and PRG2 (e.g., fatigue, comfort), it’s important to be extra cautious.
- b. Can the authors speak to why they did not use the Schaefer / Yeo atlas utilized in the prior GMV analyses for their functional connectivity analyses? This seems like it would provide the clearest structure / functional comparison.
- c. I’m not sure I follow the framing of the ROI analysis. The authors use a previously defined single ROI that corresponds to the DMN to further explore functional differences. However, they report this finding as an “increase in network coherence”. Isn’t this a comparison of beta activity within a single region – as they report a single MNI coordinate? If the authors are

indeed looking at connectivity of this ROI in relation to other regions of the DMN, this should be clarified. Otherwise, this reporting doesn't seem accurate and their use of "DMN mask" doesn't fit the traditional nomenclature of the field when referring to networks.

- i. Additionally, the choice to use this ROI from an earlier paper instead of one derived from GMV analyses in the above section is curious. Why not chose an area that showed greatest (or least) amount of overlap between PRG1 and PRG2? Why only focus on DMN, if regions functional belonging to the DAN stood out in the PRG2 – CTR comparisons?
- d. Finally, the authors appear to switch between FWE (within-network) and FDR- correction (between-network) in their fMRI analyses. Can they speak to this?

3. Third, I believe there is one more covariate that needs to be incorporated in these analyses for the PRG2 group: time since last pregnancy. Throughout the manuscript, the authors mention a partial (not full) recovery of volume in postpartum session (and the literature suggests this is relatively long lasting too). For example, they note "brain volumes slightly increased in the late postpartum period but had not returned to pre-pregnancy levels in both multiparous and primiparous women". It would be important to both describe and incorporate time since last pregnancy for the PRG2 group. Further analyses exploring GMV change within the PRG2 group alone based on time since last pregnancy could be of value.

4. Finally, my last comment relates to data quality. The authors report that each T1w image was visually checked for QC. However, given the nature of the population studied (e.g., postpartum women who may be experiencing significant fatigue), quantitative measures of T1w quality are important (e.g., Freesurfer's Euler number). Can the authors provide this? In the same vein, what was the difference in prospective data collection time between the PRG1/Control groups and the PRG2 groups – was this done at the same time?

Minor:

1. A few of the supplementary tables/figures could be clarified further. In Fig S3, what does the dashed line represent? It reads as if the PRG1 population then became the PRG2 group. Additionally, for Table S9, can the authors clarify the comparison shown here? It'd be helpful if these were color coded for strength / significance.
2. Regarding diffusion methods, can the authors clarify how the tensors were fit?
3. I believe it would be helpful for the authors to report and show % change in GMV for PRG2 just like they did in their 2022 paper. This is perhaps the most straightforward way to showcase the data and easiest for audiences to comprehend (and to benchmark against PRG1 change).

Version 1:

Reviewer comments:

Reviewer #1

(Remarks to the Author)

This is a revision of a paper that I reviewed earlier. The authors have been very responsive to detailed comments from all three reviewers, including me. They have added more detail to their results, added new correlational results linking brain change to maternal depression and attachment, and included more explanation and context. I was already favorably disposed to this paper and I believe that it now warrants publication in Nature Communications. I have no further requests or concerns.

Reviewer #2

(Remarks to the Author)

All my comments were sufficiently addressed in the revised manuscript.

Reviewer #3

(Remarks to the Author)

Thank you to the reviewers for providing a thorough and actionable response to our comments. I believe that they have adequately 1) addressed and 2) clarified the text. The new analyses, contextualizations, and additional information has made this manuscript ready for publication. I believe the scientific community, as well as the public, will be quite interested in these findings.

I have two minor follow-ups:

1. Can you also provide the % change for this PRG1 group in the Results section to make the comparisons between PRG1 and PRG2 clear and easy to digest?
2. Was age accounted for in the classifier analyses?

Great job to the authors. These data collection efforts are incredibly difficult but so valuable. It'd be amazing to see these participants tracked longitudinally through midlife!

Reviewer #1 (Remarks to the Author):

This paper examined women's brain changes after a second pregnancy, an important research question given evidence that first pregnancy appears to change the maternal brain. Thirty women undergoing a second pregnancy, 40 women undergoing a first pregnancy, and 40 control women were compared. There was overlap in terms of brain change across the first and second perinatal period, but there were also some differences, with first time mothers showing stronger default mode network changes and second time mothers showing more pronounced changes in somatomotor and dorsal attention network regions. There were also differences in resting state connectivity and white matter connectivity.

This is a very impressive study that makes a significant contribution to the research literature. I believe that the paper warrants publication, and just have a few questions or clarifications.

We thank the reviewer for his/her time and efforts to critically evaluate our paper. We appreciate the constructive comments and suggestions that helped us to improve our manuscript. We have responded to the issues/questions raised below, and addressed the specific comments raised by the reviewer as annotations to the manuscript.

First, one weakness is the lack of measures of parent-child bonding or any other individual differences measures (maternal mental health, parenting stress, amount of time spent with baby, etc). Without these measures, it is unclear whether differences between the first and second pregnancy are driven by hormones or other factors, such as parenting experience and feelings about the baby. If mothers are more bonded to their first child, do they show more or less neural change with the second pregnancy? If they feel stronger antenatal attachment during pregnancy, or stronger postnatal attachment after birth, does that explain larger magnitude of brain change? Even if there are no significant associations between self-report measures and brain changes, it would still be valuable to report null results because it would aid interpretation.

We thank the reviewer for this suggestion and agree that it is of interest to include these measures to the manuscript. We acquired questionnaire data related to maternal behaviour and mental health status during the third trimester of pregnancy and in the early postpartum period. We have now performed correlation analyses in Freesurfer using the volumetric changes across a first and second pregnancy (in comparison to CTR) with several different measures of maternal behaviour during pregnancy (Maternal Antenatal Attachment Scale (MAAS), Prenatal Attachment Inventory (PAI) and the Nesting Behaviour questionnaire) and in the postpartum period (Maternal Postnatal Attachment Scale (MPAS) and Postpartum Bonding Questionnaire (PBQ)). Additionally we performed similar correlation analyses with the Edinburgh Postpartum Depression Score and the K10 for psychological stress, both measured in the third trimester and the early postpartum period. When comparing these measures between the groups, we did not find any significant differences in maternal behaviour or mental health between the PRG1 and PRG2 groups, although we found a trend towards higher EPDS scores in the PRG2 group. Correlation analyses showed negative correlations between the volumetric change and maternal behaviour across both first and second pregnancies, except for the PBQ that showed a positive correlation. These negative correlations indicate that the more volume loss across pregnancy, the more maternal behaviour, whereas the positive correlation for the PBQ is caused by reverse scoring (lower PBQ scores represent higher maternal behaviour). The correlations are more widespread across a first pregnancy compared to a second pregnancy, especially for the PAI, MPAS and PBQ. For maternal health status, we found more widespread correlations between volumetric change and

depression and psychological distress in the PRG2 group during pregnancy, but in the PRG1 group in the early postpartum period.

We added these analyses to the results section of the manuscript in lines 142 – 167, Supplementary Fig. 6 and 7 and Supplementary Tables 3, 4 and 5. We also discussed these findings in lines 319 – 334 and lines 490 – 495, and added the information about the scales, questionnaires and analyses in the methods section in lines 725 – 754.

Was birth spacing (the length of time between the first and second pregnancies) considered in analysis? This would be interesting to measure as a continuous variable to see if brain change is greater in second-time mothers when there is more or less time elapsed after their first birth.

We thank the reviewer for this valuable suggestion. We performed a correlation analyses between the age of the first child at the PRE session, and the extracted volumetric brain change across a second pregnancy in the PRG2 group. We did not find a significant correlation for either the left or right hemisphere, suggesting that there is no difference in the amount of brain change across a second pregnancy when more or less time has elapsed since the birth of their first child.

We added these results to the results section in lines 133 – 136 and they are visualized in Supplementary Fig. 4.

The paper would also benefit for more clear explanation and interpretation of what the results mean. For example, the white matter findings are interesting but not explained clearly for readers who are not familiar with metrics such as fractional anisotropy and mean diffusivity. Do the results suggest that white matter is increasing or decreasing in volume or integrity? What does this mean in terms of brain health? Do these changes appear adaptive? In terms of the resting state findings, what are the implications of default mode network coherence? Is DMN coherence associated with stress, mental health, or empathy? Drawing on citations from the human neuroscience literature beyond maternal brain studies would help to contextualize these findings.

We agree with the reviewer that the manuscript benefits from adding more interpretation and explanation about our findings in the diffusion-weighted and resting-state data. We found a decrease in fractional anisotropy (FA) across a first pregnancy in the temporal part of the left longitudinal superior fasciculus. Although a matter of debate, lower FA values are suggested to reflect reduced integrity of white matter tracts, as the diffusion of water in the white matter is less directional with a lower FA. Previous research has linked lower FA values in the superior longitudinal fasciculus to cognitive dysfunction, for example in individuals with Schizophrenia (Peng 2020). In comparison, we found a decrease in mean diffusivity (MD) in the corticospinal tract across a second pregnancy. A decrease in MD indicates less diffusion of water molecules in all directions of the tissue, potentially reflecting an increase in white matter integrity, and has been associated with learning processes (Zatorre 2012, Tavor 2020). Additionally, as aging is associated with decreasing mean diffusivity in the brain (Bennett 2009), the effects of a second pregnancy on the MD of the corticospinal tract might oppose aging effects. Similarly, an increase in DMN network coherence as we measured across a first pregnancy is opposing the effect of aging on the brain, which is characterized by a decrease in connectivity within the DMN (Bagarinao 2019). This decrease in DMN connectivity with aging has been associated with changes in cognitive functioning (Damoiseaux 2008). On the other hand, psychiatric disorders have shared and specific phenotypes of within-network connectivity in the DMN, characterized

by both increases and decreases across various diagnosis (Doucet 2020), showing that the implications of changed DMN connectivity are not yet completely understood.

We added this information in the discussion of the manuscript in lines 304 – 310 (DMN coherence), lines 339 – 345 (reduced FA) and lines 360 – 367 (reduced MD).

I would also like to see more interpretation of some of the apparent contradictions in these findings. For example, the authors suggest that some brain changes (in the default mode, for example) are of greatest magnitude across the first pregnancy and then are fine-tuned more subtly during the second pregnancy. But if this is the case, and the brain has already done much of its restructuring in the first pregnancy, wouldn't we see baseline brain differences between first and second time mothers? Instead, it seems like everyone is starting from the same place. How do we reconcile that with the finding that the most profound brain changes seem to occur during the first pregnancy? If the brain reverts back to baseline, then presumably it would need to change in the same way to facilitate the next pregnancy. In a perfect world we could see a longitudinal study that follows the same mothers across a first and then a second pregnancy, but of course that study would be extremely challenging to conduct.

We agree that, based on our previous structural findings, we would expect to find significant baseline differences between the women undergoing a first and a second pregnancy in the areas that underwent volume reductions across a first pregnancy.

While we indeed see that the volumetric means are lower in PRG2 women in brain areas that undergo reductions across a first pregnancy, these are not statistically significantly different. This lack of observed differences in grey matter structure are likely related to the reduced power and sensitivity of cross-sectional analyses, especially because the variation in grey matter volumes at the PRE timepoint is very large. It should also be noted that the postpartum recovery process is highly dynamic, with volumetric increases shown across different time points postpartum, but also volumetric decreases from one to two years postpartum. The large range of postpartum time since the first pregnancy in our second-pregnancy group may have masked baseline differences between the PRG2 and other groups in our current study.

In lines 378 – 399, we discuss the lack of baseline effects and possible reasons underlying this finding. We have also added the suggestion of the reviewer to the text regarding future studies examining brain changes across a first and second pregnancy within the same women, which could provide a deeper insight into the recovery of brain changes in between subsequent pregnancies and the shared and distinct neural effects of successive pregnancies.

Reviewer #2 (Remarks to the Author):

This paper explores the effects of a second pregnancy on brain structure and function using multimodal MRI data. The authors report that a second pregnancy conferred similar though less pronounced changes in the default mode and frontoparietal network, yet stronger changes in the dorsal attentional network and somatomotor network compared to a first pregnancy. The current literature lacks a detailed characterization of how first versus second pregnancies influence brain health, and this paper addresses this gap nicely. This is a high-quality and ambitious study that tackles a critical yet underexplored question. The prospective pre-conception design is a major strength and adds valuable insight. Overall, this paper would make a substantial contribution to

the literature on maternal brain plasticity. I have several comments to help temper interpretations of the findings and to acknowledge potential limitations.

We thank the reviewer for his/her time and efforts to critically evaluate our paper. We appreciate the constructive comments and suggestions that helped us to improve our manuscript. We have responded to the issues/questions raised below, and responded within the manuscript to the specific comments raised by the reviewer as annotations to the manuscript.

The introduction is strong and clearly written. However, it is somewhat vague in its description of the effects of multiple pregnancies in both animal and human studies. I recognize there are limited papers examining the effects of previous pregnancies on brain health, but the authors should consider citing additional recent work and outlining the more specific differences between differing levels of parity. In addition, several human studies have assessed parity as a linear/continuous variable, offering relevant insight into how additional pregnancies may shape brain health.

We thank the reviewer for this valuable comment and added some more literature regarding the long-term effects of pregnancy on the brain in rodents (lines 54 – 56) and also added more literature on how parity may influence brain health and cognition, and the risk for and effects of other brain disorders in human studies in lines 62 – 69.

Please provide a clearer justification for why the specific neural metabolites were selected for analyses. This would help contextualize the results and clarify their relevance to the broader literature on pregnancy-related brain changes.

We thank the reviewer for pointing this out. We estimated metabolite concentrations using LCMoDel, using a dataset containing in total seventeen metabolites. Of these metabolites, we considered just the reported metabolites (tNAA, tCr, tCho, Glu and Ins), as these metabolites (or combinations thereof) can typically be measured with high precision, indicated by their Cramer-Rao lower bounds well below 10%.

This information is also described in lines 703 – 704.

Many studies that have examined the effects of parity on brain health have focused on short-term outcomes. It would strengthen the discussion to explicitly distinguish between short- and long-term effects of pregnancy on the brain. The authors briefly mention longer-term signatures of pregnancy on brain plasticity in humans. However, it would be valuable to integrate evidence from rodent models describing how previous pregnancies influence brain plasticity at middle age to provide a more comprehensive perspective.

We thank the reviewer for pointing this out. We agree that adding information about long-term effects of parity in rodent studies helps to interpret our findings. Therefore, we added this information in the discussion of the manuscript in lines 406 – 412.

The reported 80% classification accuracy between PRG1 and PRG2 is promising. However, given the relatively small sample size, there is a risk of overfitting. Please clarify whether any cross-validation procedures were used and discuss this limitation more explicitly.

We apologize for the unclarity. We have performed a leave one subject out cross-validation for the performed classification analyses, due to the relatively low sample size of our classes, making the training dataset for the classifier as large as possible. However, we do agree with the reviewer that the relatively small sample size also means a risk of overfitting, which we now discuss in the discussion in lines 442 – 451. We validated our cross-validation method by repeating our classification analyses using a k-fold cross validation with 10 folds, to reduce the risk of overfitting. Although the accuracy went down slightly as expected, the classifications between all combinations of groups were still significant. We also describe the results of this alternative way of cross-validation in our manuscript in lines 126 – 130 and describe the method in lines 592 – 596.

Additionally, the manuscript would benefit from a more detailed discussion of potential confounding variables in the context of pregnancy effects, including interpregnancy interval, age of first pregnancy, breastfeeding, and other relevant factors.

We included a correlation analyses between the volumetric change across a second pregnancy and the interpregnancy interval (age of first child at the PRE session in PRG2), and showed that the interpregnancy interval is not statistically significantly correlated to the volumetric changes across a second pregnancy. We added this analysis to the manuscript in lines 133 – 136, and the results are visualized in Supplementary Fig. 4.

We acknowledge that other potential confounding factors, such as age of first pregnancy, breastfeeding status, type of delivery, social support and sleep may have influenced our observed brain changes across a second pregnancy. Although our previous findings suggest that several of these factors such as sleep, stress, and breastfeeding do not strongly contribute to pregnancy-induced brain changes, these only represent a subset of potentially relevant variables and it is likely that such factors are associated with more subtle effects that only surface in larger samples. Therefore, studies involving larger samples that track these measurements are likely necessary to be able to acquire more insights into the influence of such factors on pregnancy-induced brain changes. In the revised manuscript, we now discuss the influence of potential confounding variables in lines 466 – 475.

Reviewer #3 (Remarks to the Author):

Thank you for inviting me to review the manuscript entitled “The effects of a second pregnancy on women’s brain structure and function” by Straathof and colleagues. This study examines a remarkably rare and valuable cohort, and the questions it addresses are of clear significance to the field. Much of the prospective work to date—most notably Dr. Hoekzema’s groundbreaking studies—has focused on women experiencing their first pregnancy. While these investigations have been foundational, they overlook the fact that most women will experience multiple pregnancies in their lifetime. What happens to the brain in subsequent pregnancies? This study begins to address that critical gap.

We thank the reviewer for his/her time and efforts to critically evaluate our paper. We appreciate the constructive comments and suggestions that helped us to improve our manuscript. We have responded to the issues/questions raised below, and responded within the manuscript to the specific comments raised by the reviewer as annotations to the manuscript.

However, I have several conceptual and methodological concerns that I would like to see addressed and clarified before I can recommend publication. I have outlined both major and minor considerations below.

1. My first concern involves the consistency in reporting throughout the Results section, as well as the conceptual meaning behind these findings.

a. The authors find no significant differences when comparing vertex-wise GMV volume between the PRG1 and PRG2 groups, but visual inspection triggered a follow-up analysis (classification of GMV changes) that showed different takeaways. Can the authors speak to how we should reconcile this: are there true differences between PRG1 and PRG2, and/or is it based on the methodological approach?

We thank the reviewer for mentioning this critical point. We noticed that pregnancy, being either a first or a second, has very large effects on brain volumes, so that the more “subtle” differences, such as differences between a first and second pregnancy, are difficult to pick up. We calculated the effect sizes (as Cohen’s D) for the vertex-wise analyses of all three contrasts (CTR – PRG2, CTR-PRG1 and PRG1-PRG2), and found large to very large effect sizes for the comparisons between the control women and both pregnancy groups (Cohen’s D > 1.0), while we found moderate to large effect sizes for the comparison between PRG1 and PRG2 (Supplementary Table 1). In order to also allow for slightly less prominent effects to surface, we have now also applied a less stringent method to correct for multiple testing (permutation testing using a vertex-wise threshold of $p < 0.01$, 1000 permutations, and correcting for measuring across both hemispheres) for the PRG1-PRG2 comparison specifically, and found areas in the vertex-wise analyses that significantly changed differently over time across a first and second pregnancy (Supplementary Figure 2). Areas with larger changes across PRG1 (shown in blue in Supplementary Figure 2) were located in the default mode network, whereas areas with larger changes across a second pregnancy (shown in red) were mainly located in the dorsal and ventral attention network and sensorimotor network. This is in line with our subsequent analyses focussing on changes in the different networks. In addition, our classification approach also supports our observation of different brain changes in PRG1 and PRG2.

We added the results of these analyses in the manuscript in lines 105 – 115 and Supplementary Figure 2 and Supplementary Table 1. Additionally, we described the methods used in lines 574 – 579.

b. It’s not always clear when the authors are looking at POST1 in both or just the PRG2 group, and why those changes are not incorporated in all analyses. For example, it looks like POST1 sessions were explored with respect to GMV trajectories, but not incorporated into the classifier?

We apologize for the confusion. We used the POST1 timepoint to study the longer-term effects of a second pregnancy. However, since we only had this POST1 timepoint in a small subset of women ($n=14$), we only included it in analyses where we found a significant pregnancy effect from PRE to POST in PRG2. Therefore, it is included in the vertex-wise volumetric analyses and the diffusion-weighted analyses. The results of the POST1 timepoint across a first pregnancy (PRG1) are discussed in our previously published manuscript (Hoekzema 2022). They were also not included in the classifier, as the sample size for the classification analyses was already relatively small, and this would have reduced it even more.

We added this clarification in the manuscript in lines 512 – 514.

2. My second concern involves the nature of the resting-state data. The authors are using incredibly underpowered RS data - 137 images per session at a TR > 2 seconds – to conduct functional brain network analyses (cf. Birn NeuroImage, 2013). I believe they are too underpowered to do this reliably and this section should be preferably 1) removed entirely or 2) moved to supplement & addressed as a significant limitation. If the latter:

We thank the reviewer for this comment. We agree with the reviewer that our resting-state data is not of the highest possible quality. This approach was chosen due to strict time limitations and the difficult population to scan. It should also be noted that this was the first study to apply any form of fMRI measurements in such a cohort, and that this study started over 10 years ago when 5 minutes for resting state acquisitions were still quite standard. Nevertheless, we still believe the data is valuable as a comparison to our earlier published manuscript reporting on changes in resting-state functional connectivity across a first pregnancy, with the same MRI protocol. Therefore, we moved the results of the resting-state analyses to the supplement (Supplementary Fig. 8 and Supplementary Tables 13 – 15), and added information about the quality of the resting-state fMRI to the discussion in lines 452 – 459. We also respond to the raised concerns about the resting-state fMRI data analyses point by point below.

a. Can the authors also provide a breakdown of head motion (e.g., FWD) for the different groups? Given the nature of the postpartum sessions in PRG1 and PRG2 (e.g., fatigue, comfort), it's important to be extra cautious.

*We thank the reviewer for this important point. We determined the mean frame-wise displacement for the different groups (PRG1, PRG2 and CTRL) at the PRE and POST timepoint. We did not find any statistically significant group*session interaction effects, indicating that there is no difference in the head motion for the different groups at the different time points. We included this information in the manuscript in lines 620 – 623 and Supplementary Table 22.*

b. Can the authors speak to why they did not use the Schaefer / Yeo atlas utilized in the prior GMV analyses for their functional connectivity analyses? This seems like it would provide the clearest structure / functional comparison.

We used the Yeo atlas for our volumetric analyses as this atlas is based on surface-based clustering methods in Freesurfer (Yeo 2011), and hereby might best reflect the structural correlates of functional networks. They assume in their paper introducing the atlas that functional connectivity reflects stable properties of cortical architecture. Since we used an ICA analysis to identify functional networks in our resting-state fMRI data, we used the Smith 2009 atlas which is also based on an ICA approach defining networks by their spatiotemporal configuration and functional roles. Using this ICA-based atlas, we expected the most representative functional networks in our resting-state analyses. We added this in line 628 in the manuscript.

c. I'm not sure I follow the framing of the ROI analysis. The authors use a previously defined single ROI that corresponds to the DMN to further explore functional differences. However, they report this finding as an "increase in network coherence". Isn't this a comparison of beta activity within a single region – as they report a single MNI coordinate? If the authors are indeed looking at connectivity of this ROI in relation to other regions of the DMN, this should be clarified. Otherwise,

this reporting doesn't seem accurate and their use of "DMN mask" doesn't fit the traditional nomenclature of the field when referring to networks.

We apologize for the unclarity about the ROI analyses. To determine within-network connectivity, we determined the connectivity for each voxel in the network with all the other voxels inside the specific network, and averages this across the entire network. In line with this, instead of using all voxels in the DMN, we additionally performed an ROI analysis, using the DMN mask that showed a change in within-network coherence across a first pregnancy. We performed a within-network correlation analyses by determining the connectivity of each voxel inside this ROI, with all the other voxels within the DMN. We also clarified this in the manuscript in lines 630 – 631 and 643 – 645.

i. Additionally, the choice to use this ROI from an earlier paper instead of one derived from GMV analyses in the above section is curious. Why not chose an area that showed greatest (or least) amount of overlap between PRG1 and PRG2? Why only focus on DMN, if regions functional belonging to the DAN stood out in the PRG2 – CTR comparisons?

We apologize for the confusion, but we did not solely focus on the DMN in our resting-state fMRI analyses. We identified the ten networks of Smith 2009 and determined the within-network connectivity within these networks, and the between-network connectivity between all combinations of these networks. These analyses did not render significant results (see Supplementary Tables 13-15). As a follow-up analysis, we used the DMN mask from our previous manuscript (Hoekzema 2022) that represented the area of change in DMN coherence across a first pregnancy, and calculated the within-network connectivity of the voxels inside this mask with the rest of the DMN, to identify whether the changes that occur across a first pregnancy also take place across a second pregnancy.

d. Finally, the authors appear to switch between FWE (within-network) and FDR- correction (between-network) in their fMRI analyses. Can they speak to this?

We apologize for the confusion. This difference in the method to correct for multiple testing is due to limitations of the software packages used, and types of analyses performed. For within-network analyses, we used SPM12, and used a FWE-correction across all voxels. For the between-network analyses we used an FDR-correction in Rstudio as these included less tests and FWE-corrections are not standard in Rstudio.

3. Third, I believe there is one more covariate that needs to be incorporated in these analyses for the PRG2 group: time since last pregnancy. Throughout the manuscript, the authors mention a partial (not full) recovery of volume in postpartum session (and the literature suggests this is relatively long lasting too). For example, they note "brain volumes slightly increased in the late postpartum period but had not returned to pre-pregnancy levels in both multiparous and primiparous women". It would be important to both describe and incorporate time since last pregnancy for the PRG2 group. Further analyses exploring GMV change within the PRG2 group alone based on time since last pregnancy could be of value.

We thank the reviewer for this valuable suggestion. We performed a correlation analyses between the age of the first child at the PRE session, and the volumetric change across a second pregnancy in the PRG2 group. We did not find a significant correlation for either the left or right hemisphere,

suggesting that there is no difference in the amount of brain change across a second pregnancy when more or less time has elapsed since the birth of the first child.

We added this analysis to the manuscript in lines 133 – 136, and the results are visualized in Supplementary Fig. 4.

4. Finally, my last comment relates to data quality. The authors report that each T1w image was visually checked for QC. However, given the nature of the population studied (e.g., postpartum women who may be experiencing significant fatigue), quantitative measures of T1w quality are important (e.g., Freesurfer's Euler number). Can the authors provide this? In the same vein, what was the difference in prospective data collection time between the PRG1/Control groups and the PRG2 groups – was this done at the same time?

*We are grateful to the reviewer for this comment, and appreciate the suggestion to include Freesurfer's Euler number as a measure of data quality of our T1w images. We determined the Euler number for each individual T1w image, and calculated the average Euler number per group per time point, which we added in Supplementary Table 21. The mean Euler numbers are lower than Freesurfer's own exemplar dataset BERT dataset as described in Chalavi et al., 2012. We did not find a statistically significant group (PRG1, PRG2, CTR) * session (PRE, POST, POST1) interaction effect, which is described in the manuscript in lines 536 – 540. Additionally, after processing in Freesurfer all T1w scans had an Euler number of 2, which represents an accurate reconstruction.*

Additionally, the data for the different groups was collected at the same time, meaning that subjects of the different groups were scanned intertwined, which we also added to the manuscript in lines 519 – 520.

Minor:

1. A few of the supplementary tables/figures could be clarified further. In Fig S3, what does the dashed line represent? It reads as if the PRG1 population then became the PRG2 group. Additionally, for Table S9, can the authors clarify the comparison shown here? It'd be helpful if these were color coded for strength / significance.

We agree with the reviewer that the dashed line in Supplementary Fig. 3 (old version, now Supplementary Fig. 5) is confusing and reads as if the same population has been scanned across a first and second pregnancy. Therefore, we removed the dashed line.

*Additionally, we clarified what is shown in Supplementary Table 9 and 10 (old version, now Supplementary Table 13 and 14). These tables show the results of the group*session interaction effect for between-network connectivity between each combination of our identified resting-state networks, with a comparison between PRG2 and CTR in Supplementary Table 9 (now Supplementary Table 13) and between PRG2 and PRG1 in Supplementary Table 10 (now Supplementary Table 14). We also color-coded the tables according to the F-values of the effect, with higher F-values represented by darker red colours.*

2. Regarding diffusion methods, can the authors clarify how the tensors were fit? *We apologize for the unclarity, we fitted the diffusion tensors using standard dwi2tensor in*

MRtriX3, and described the fitting of the tensors in more detail in the manuscript in lines 666 – 669.

3. I believe it would be helpful for the authors to report and show % change in GMV for PRG2 just like they did in their 2022 paper. This is perhaps the most straightforward way to showcase the data and easiest for audiences to comprehend (and to benchmark against PRG1 change).

We thank the reviewer for this valuable comment, and we calculated the %change in GMV for the PRG2 group. The median percentage volume change across a second pregnancy (in regions that significantly differed in the PRG2 vs CTR contrast) was 2.8%, which we added in the manuscript in lines 95 – 96.

Response to Reviewers

We would like to thank the reviewers for their time and their careful evaluation of our manuscript. We are very glad that we were able to address the raised concerns and believe it has allowed us to significantly improve the quality of our work. Below we respond to the minor follow-ups requested by reviewer 3. We have adjusted the manuscript to address these comments as well as all editorial requests outlined in the Author Checklist.

Reviewer #1 (Remarks to the Author):

This is a revision of a paper that I reviewed earlier. The authors have been very responsive to detailed comments from all three reviewers, including me. They have added more detail to their results, added new correlational results linking brain change to maternal depression and attachment, and included more explanation and context. I was already favorably disposed to this paper and I believe that it now warrants publication in Nature Communications. I have no further requests or concerns.

Reviewer #2 (Remarks to the Author):

All my comments were sufficiently addressed in the revised manuscript.

Reviewer #3 (Remarks to the Author):

Thank you to the reviewers for providing a thorough and actionable response to our comments. I believe that they have adequately 1) addressed and 2) clarified the text. The new analyses, contextualizations, and additional information has made this manuscript ready for publication. I believe the scientific community, as well as the public, will be quite interested in these findings.

I have two minor follow-ups:

1. Can you also provide the % change for this PRG1 group in the Results section to make the comparisons between PRG1 and PRG2 clear and easy to digest?

We have now also computed this for the first-time mothers and added this to the Results section of the revised manuscript (line 92 of the revised manuscript).

2. Was age accounted for in the classifier analyses?

These analyses did not include age as a covariate, we had overlooked this, our apologies. We have now performed these analyses including age as a regressor (with both cross-validation approaches), which rendered highly similar results. These are now reported in lines 123-127 of the Results section of the revised manuscript.

Great job to the authors. These data collection efforts are incredibly difficult but so valuable. It'd be amazing to see these participants tracked longitudinally through midlife!

We thank the reviewer for these very kind words.